# Nutritional Support in Patients with Severe Acute Pancreatitis-Current Standards

**DOI:** 10.3390/nu13051498

**Published:** 2021-04-28

**Authors:** Beata Jabłońska, Sławomir Mrowiec

**Affiliations:** Department of Digestive Tract Surgery, Medical University of Silesia, Medyków 14 St., 40752 Katowice, Poland; mrowasm@poczta.onet.pl

**Keywords:** nutritional support, enteral nutrition, parenteral nutrition, severe acute pancreatitis, immunonutrition, glutamine

## Abstract

Severe acute pancreatitis (SAP) leads to numerous inflammatory and nutritional disturbances. All SAP patients are at a high nutritional risk. It has been proven that proper nutrition significantly reduces mortality rate and the incidence of the infectious complications in SAP patients. According to the literature, early (started within 24–48 h) enteral nutrition (EN) is optimal in most patients. EN protects gut barrier function because it decreases gastrointestinal dysmotility secondary to pancreatic inflammation. Currently, the role of parenteral nutrition (PN) in SAP patients is limited to patients in whom EN is not possible or contraindicated. Early versus delayed EN, nasogastric versus nasojejunal tube for EN, EN versus PN in SAP patients and the role of immunonutrition (IN) in SAP patients are discussed in this review.

## 1. Introduction

### 1.1. Definition and Epidemiology of AP

Acute pancreatitis (AP) is an inflammatory disease involving a pancreatic parenchyma and peripancreatic tissues with a potential systematic immune response in a severe disorder course. The incidence of AP has increased in most countries. The mean global AP incidence is 34/100,000 [1]. The incidence of AP in Europe ranges from 4.6 to 100/100,000 [2]. Poland is one of the countries with the highest incidence rate—72.1/100,000 [3].

### 1.2. Classification of AP

The revised Atlanta classification (2012) [4] distinguished interstitial edematous (1) and necrotizing (2) acute pancreatitis types. In most patients, the first AP type is observed but, in 5–10% of cases, necrotizing AP occurs. According to the updated Atlanta classification, necrotizing AP most frequently involves the whole pancreas and peripancreatic tissues, less frequently only the peripancreatic tissues, and rarely only the pancreas without adjacent tissues [4]. The authors of the Atlanta classification distinguished two overlapping AP phases: early and late. The early phase usually lasts for the first week and a second delayed phase can be prolonged to weeks or months [4]. In the early phase, the local pancreatic inflammation leads to generalized immune alterations as a response to the pancreatic injury. The pancreatic inflammatory injury triggers a cytokine storm which is demonstrated as the systemic inflammatory response syndrome (SIRS). The prolonged SIRS can lead to the multiorgan dysfunction syndrome (MODS) which determines the AP severity. In the second delayed phase, systemic and/or local complications are observed. This phase is noted only in patients with moderately severe or severe acute pancreatitis [4].

### 1.3. Pathomechanism of AP, the Role of Oxidative Stress in AP

In AP, the secretion of digestive enzymes is disturbed. Digestive enzymes are transported outside the cells. Additionally, in a damaged pancreas, the separation of digestive enzymes from lysosomal hydrolases is disturbed. Therefore, both types of enzymes are colocalized within intracellular vacuoles. This colocalization phenomenon causes the premature activation of digestive enzymes. The subsequent rupture of these vacuoles liberates the activated digestive enzymes in the cytoplasmic space, followed by a cascade of events leading to AP. The activation of pancreatic enzymes in and around the pancreas and bloodstream leads to pancreatic coagulation necrosis as well as necrosis and hemorrhage of peripancreatic and peritoneal adipose tissue. Oxidative stress plays an important role in the early phase of AP [5]. Moreover, in AP, the release of proinflammatory cytokines and mediators and the increased intestinal permeability and is noted. Intestinal barrier dysfunction results in infected necrosis, bacteremia and multiorgan failure. Therefore, support of the alimentary tract is very important in the AP treatment. Stress increases the inflammatory infiltration within the intestine, increases damage to the intestinal barrier, and allows bacterial translocation into the bloodstream. Therefore, antioxidants should be helpful for AP treatment [5]. Some antioxidants (such as resveratrol, selenium, ascorbic acid, melatonin, hydroxythyrozol, cerulein), have been investigated as potential beneficial agents in AP patients, but these investigations have mainly involved animal models, not human participants [6].

### 1.4. Assessment of the Severity of AP

Management in AP depends on the severity of the disease. Therefore, identification of patients with potentially severe acute pancreatitis (SAP), who need nutritional support (NS), is very essential. The authors of the Atlanta classification defined and stratified the severity of AP. Three types of AP severity have been identified: mild acute pancreatitis (MAP), moderately severe acute pancreatitis (MSAP), and severe acute pancreatitis (SAP). Severity grading of acute pancreatitis according to the revised Atlanta classification is presented in Table 1. The presence of organ deficiency and local or systemic complications differs the above mentioned AP types. The organ failure it is not reported in MAP, it is transient in MSAP, and it is persistent (>48 h) in SAP. It can include single organ failure or multiple organ failure. These patients require aggressive treatment including nutritional intervention [4].

Assessment of AP severity is crucial for appropriate management. Several single parameters or more complex scores have been developed to assess AP severity. The Ranson score is the oldest score for the prediction of AP severity. It involves 11 parameters including 5 parameters assessed at admission (scores 0–5), and 6 parameters—48 h later (scores 0–6). Although a score ≥3 has a high sensitivity and specificity regarding a severe course of AP (83.9% and 78.0%, respectively) and a negative predictive value of 94.5%, the severity can be predicted no earlier than 48 h after admission. The later introduced scores for the prediction of AP severity are as follows: the Glasgow (Imrie) score (8 parameters), Multiple Organ Dysfunction Score (MOSS/MODS) (12 parameters), Bedside Index of Severity of Acute Pancreatitis (BISAP) score (5 parameters), and the Acute Physiology and Chronic Health Evaluation (APACHE II) score (14 parameters). All the scores include various clinical and laboratory parameters. The sensitivity and specificity of these scores for predicting SAP is 55–90%. The inability to obtain a complete score until at least 48 h of the disease is a limitation of the Ranson and Glasgow scores, while the complexity of the scoring system itself, is a limitation of the APACHE II score (including age, Glasgow coma score, vital and oxygenation parameters, as well as biochemical and hematological parameters). The APACHE II score was originally developed to predict mortality in intensive care patients. The APACHE II score ≥ 8 predicts severe acute pancreatitis (sensitivity of 65–83%, and specificity of 77–91%) [7]. BISAP score (including blood urea nitrogen (BUN), impaired mental status, the systemic inflammatory response syndrome (SIRS), age and pleural effusion) allows to predict the AP severity within the first 24 h of hospitalization [7].

C-reactive protein (CRP) is the most useful single laboratory parameter. Despite its delayed increase (peaking not earlier than 72 h after the onset of AP) it is accurate and widely available. An elevated CRP level > 150 mg/L is a prognostic factor for complicated course (sensitivity of 85%) in the first 72 h after the onset of symptoms [8,9]. The other biochemical parameters (albumin, creatinine, procalcitonin) can be useful in the prediction of SAP. While their predictive value is unsatisfactory when used alone, their use in combination can effectively improve the sensitivity of prediction. Therefore, the CRP/albumin ratio and creatinine/albumin ratio have been reported to be useful in prediction of SAP [9].

Premature intra-acinar activation of trypsinogen leads to the cytokine storm in AP. Therefore, various cytokines have been tested to be useful for prediction SAP. Among, them, proinflammatory interleukin-6 (Il-6) is the most useful parameters. The plasma level of IL-6 is elevated early phase of AP. It is a sensitive and specific marker to predict organ failure and SAP [10]. A few studies have shown that elevated levels of the tissue necrosing factor-α (TNF-α) predicted SAP [11,12]. The opinions regarding the utility of the TNF-α in prediction SAP are contradictory. According to another authors, it is elevated only in a small percentage (9–36%) of SAP patients, and it cannot be used as a prognostic factor in AP [10,13,14]. Similarly, the significance of IL-10 regarding severity of AP is yet not clear because of the contradictory reports [10,11,15]. The other cytokines that have been assessed for predicting AP are as follows: IL-8, and IL-18 [8].

Angiopoietin-2 (Ang-2), a type of glycoprotein acting selectively on endothelial cells and leading to an increase in endothelial permeability, has been identified to be a useful predictor for acute gastrointestinal injury and intestinal barrier dysfunction in patient with AP [16].

The morphologic severity of AP can be determined using a CT severity index (CTSI) that was developed by Balthazar and then modified and extended to monitor organ failure by Silverman, Banks, and colleagues in 2004 [8].

The following cutoff values positively correlate with the severity grade of AP: Ranson ≥ 3, Glasgow ≥ 3, MOSS ≥ 5, BISAP ≥ 2, APACHE II ≥ 8, CTSI ≥ 5, procalcitonin ≥ 0.5 ng/mL, CRP ≥ 150 mg/L, and IL-6 ≥ 50 pg/mL [8].

### 1.5. Disturbances of the Nutritional Status in SAP

SAP is a serious acute disease. In these patients, resting energy expenditure (REE) increases due to inflammation-induced hypermetabolism and/or septic complications. In SAP, protein catabolism and increased energy requirements are observed. The disease leads to undernutrition and disturbances of water/electrolyte and acid-base balance. Therefore, energy and protein requirements are higher in patients with SAP [17,18,19]. The pancreatic damage and infectious complications are associated with hyperglycemia which must be controlled, is induced by damage to beta cells, insulin resistance, and infectious complications. Therefore, insulin therapy should be considered in nutritional support in patients with SAP. Acute pancreatitis is usually associated with deficiency of numerous vitamins and micronutrients such as vitamins B1, B2, B3, B12, C, A, folic acid, and zinc. Moreover, dysregulations in water/electrolytes and of acid-base balance are noted in SAP. Hypocalcemia is observed in 40–60% of patients due to saponification of calcium. The other disturbances are as follows: hypomagnesemia, decreased parathyroid hormone release, and increased calcitonin levels, hypertriglyceridemia [19]. The level of, must be closely monitored during parenteral nutrition including lipid emulsions. It has been noted that protein loss with a negative nitrogen balance occurred in 80% of patients with SAP [3]. The nitrogen loss can be up to 20–40 g/d. Negative nitrogen balance is associated with increased mortality [20].

Therefore, all patients with SAP are at nutritional risk and require nutritional intervention [17]. Moreover, just at the SAP beginning, damage to enterocytes due to microcirculatory injury and gut hypoperfusion leads to increased gastrointestinal permeability and translocation of systemic cytokines, toxins, and bacteria [18,20]. Therefore, the goal of the nutritional support (NS) in AP is not only to prevent and treat malnutrition, but also modulate and decrease altered inflammatory response [21]. Nutrition constitutes the third step of AP management according to the “PANCREAS” (perfusion, analgesia, nutrition, clinical and radiological assessment, endoscopy, antibiotics, surgery) acronym described by Khaliq et al. [22]. This acronym is very useful in management with SAP patients [3,5]. In Khaliq’s et al. [22] report, enteral nutrition within 48 h (with or without the use of nasojejunal tube) led to reduction of mortality in SAP patients [22].

### 1.6. Comparison of Management and Nutritional Support in Acute and Chronic Pancreatitis

The overall management including nutritional support is generally different in patients with AP and chronic pancreatitis (CP), but there are also some similarities. Regarding general management, in both AP and CP, conservative treatment plays the most essential role in most patients, and surgery is reserved for patients with complications of these diseases. The complication types different in both groups. The infected necrosis, abscess, ileus, gastrointestinal perforation, hemorrhage are typical indications for surgery in AP patients. In CP, indications for surgery are the following: duodenal or bile duct stenosis, pancreatic tumor, pancreatic lithiasis with a Wirsung duct dilatation and small duct pancreatitis with an intractable pain. Regarding nutritional support, in MAP, the early oral low-fat diet is recommended, while in SAP, enteral feeding is used. Parenteral nutrition is reserved only for patients with intolerance or impossibility of enteral nutrition. The majority of patients with CP can receive normal food with supplementation of exogenous pancreatic enzymes. Generally, 10–15% of patients need oral nutrition supplementation and 5% require tube feeding. Oral elemental supplements are associated with a significant decrease pain and an improvement in nutrition indices. Additionally, fat-soluble (A, D, E, K) and water-soluble (vitamin B12, folic acid, thiamine) vitamins as well as minerals such as magnesium, iron, selenium and zinc should be administered in cases of confirmed deficiency. Parenteral nutrition is used in <1% of patients with complication of CP such as duodenal stenosis, complex fistula, or malnourished patients with gastrointestinal dysfunction prior to surgery. Similar to nutrition in AP, oral and enteral routes of nutrition are preferable, and parenteral nutrition is reserved for patients in whom oral and enteral nutrition is not possible [23,24].

The role of supplementation of pancreatic enzymes is different in AP and CP patients. In patients with AP, these supplements are not generally recommended except for obvious pancreatic exocrine insufficiency (PEI). The information regarding the occurrence of PEI in AP patients during hospitalization is not sufficient. PEI can be caused by the pancreatic necrosis, but it is manifested following AP [25]. The knowledge of PEI after AP is better. PEI is noted at least during the first 6 to 18 months after AP, and the grade of dysfunction is related to the AP severity [26]. PEI is common in patients with CP. The low-fat oral diet in MAP, or special formula of enteral/parenteral nutrition in SAP (when oral diet is not possible) are commonly used in most patients. In CP, supplementation of exogenous pancreatic enzymes is very essential due to maldigestion and malabsorption in the disease course [27]. A large meta-analysis by Hollemans et al. [28], containing 32 studies (1495 patients), showed a 27.1% prevalence of PEI after AP. It was positively correlated with degree of pancreatic parenchymal injury, alcoholic cause (22.7%), SAP (33.4%), and necrotizing pancreatitis (32%). Therefore, clinical observations in terms of symptoms of maldigestion (diarrhea, steatorrhea) and/or non-invasive pancreatic function investigations (such as fecal fat and fecal elastase) for at least 6–18 months after AP, especially in alcoholic, severe, and necrotizing pancreatitis are recommended. In clinically relevant PEI, supplementation of pancreatic enzymes is recommended [19,28].

Nutritional support is very essential to correct nutritional status and modify altered immune response in SAP patients. Some aspects of nutritional support, regarding the optimal timing, type (enteral nutrition (EN) vs parenteral nutrition (PN), via nasogastric tube (NGT) or nasojejunal tube (NJT)), and the role of immunonutrients, such as glutamine (Gln), Arginine (Arg), polyunsaturated omega-3-fatty acids (PUFA), nucleotids in nutrition of patients with SAP, are discussed.

## 2. The Literature Searching and Review

We have reviewed PubMed database. The search terms and mesh heading were as follows: “acute pancreatitis”, or “severe acute pancreatitis”, and “nutrition” or “nutritional support” or “nutritional intervention” or “enteral nutrition” or “enteral feeding” or “nasogastric tube” or “nasojejunal tube” or “nasoenteric tube” or “parenteral nutrition” or “immunonutrition” or “immunomodulating nutrition” or “glutamine” or “omega-3-fatty acids”. The selected articles were discussed.

## 3. European Society for Clinical Nutrition and Metabolism (ESPEN), American Gastroenterological Association (AGA), and UK Guidelines on Clinical Nutrition in Severe Acute Pancreatitis

ESPEN, AGA, and UK guidelines recommend the early oral nutrition in MAP patients and enteral nutrition in SAP patients with impossibility of oral feeding. Parenteral nutrition should be reserved for patients in whom enteral nutrition is not possible or not tolerated. According to ESPEN guidelines, the nasogastric tube is preferred over the nasojejunal tube, and the nasojejunal tube should be reserved for patients with gastric outlet obstruction. Also, according to UK guidelines, enteral feeding via nasogastric tube should be effective in 80% of patients. AGA recommends using either a nasogastric or nasojejunal tube. Regarding optimal timing of starting nutrition, ESPEN guidelines recommend starting nutrition within 24–72 h of admission. According to AGA, early feeding is not successful in all AP patients due to pain, vomiting, or ileus, and feeding may need to be delayed beyond 24 h in some cases [27,29,30].

The ESPEN authors recommend the use of parenteral glutamine in PN and they do not recommend immunonutrition in other cases in SAP. There is no information regarding the use of glutamine and immunonutrition in AGA recommendations. According to UK guidelines, there is no sufficient reports to recommend standard immune enhanced formulations. The probiotics and a supplementation of pancreatic enzymes are not recommended by ESPEN. There is no information on probiotics and pancreatic enzyme supplementation in AGA and UK guidelines [27,29,30,31].

## 4. Gut Rousing, But Not Resting, and No “Pancreatic Rest” in SAP Patients

Historically, a “pancreatic rest” and “bowel rest” with “nil per os” (NPO) or “nothing per mouth” status were widely postulated. Previously, it has been thought that feeding administered into the gastrointestinal tract proximal to mid-jejunum (around 40 cm distal to ligament of Trietz) increases the secretion of pancreatic enzymes. It might be associated with a pancreatic autodigestion leading to deterioration of the clinical AP course. Therefore, it has been thought in order to decrease pancreatic secretion, EN should not be used and PN should be administered for “pancreatic rest”. Currently, it is known that in AP patients, pancreatic exocrine function deteriorates proportionally to disease severity with significantly decreased pancreatic secretion observed in SAP patients with pancreatic necrosis. This fact suggests that pancreatic exocrine function is significantly altered in AP. Therefore, EN, regardless the route, especially in SAP, should not increase pancreatic secretion. The discovery of this phenomenon became a breakthrough in the nutritional management of SAP patients [19,32]. In addition, numerous studies have shown that EN is safe and effective in AP patients and is associated with lower mortality, organ failure, infectious complications and surgery rates [19,33,34,35,36,37]. PN is not sufficient in SAP patients, because can lead to the increased gut dysfunction without trophic nutrition necessary for enterocytes and gut integrity, because it has been shown that exclusive PN can lead to the intestinal atrophy [18].

## 5. The Optimal Route of Nutritional Support in SAP Patients: Enteral versus Parenteral Nutrition

As we mentioned above, it has been shown that the enteral route is optimal for NS in SAP patients. Apart mentioned above advantages, EN is safe and less expensive compared to PN [38]. There are numerous reports which confirm this theory in the world literature. The results of the most important studies are presented in Table 2 [39,40,41,42,43].

It is known that gastrointestinal dysmotility can lead to EN intolerance in SAP patients. EN increases gastrointestinal motility and has trophic impact on enterocytes [44]. It is important to continue EN in SAP despite the gut dysmotility secondary to abdominal compartment syndrome (ACS) due to ascites [45]. Hongyin et al. [45] analyzed the impact of abdominal paracentesis drainage (APD) on the possibility of enteral feeding in AP patients. This study included 161 AP patients hospitalized between January 2015 and April 2016. The patients were divided into two groups: the APD group and the non-APD group. This study showed that APD might enable the supply of enteral feeding in AP patients [45].

All presented articles have shown the superiority of EN in SAP patients. According to most authors, EN is associated with a lower morbidity and mortality rates, shorter duration and lower cost of hospitalization. Summary of articles regarding comparison of EE and PN in SAP is presented in Table 2.

We fully agree with these opinions, and also prefer EN in nutritional support in SAP patients. The all above mentioned articles were summarized in Table 2.

## 6. The Optimal Timing of Nutritional Support in SAP Patients

It is known that immune dysregulation secondary to the cytokine storm is observed in SAP patients. Therefore, currently, there is an opinion that early enteral nutrition (EEN) could increase antioxidant activity, modulate inflammatory response, and decrease the risk of MODS [45]. Summary of articles regarding comparison of early and delayed enteral nutrition in SAP is presented in Table 3.

According to most authors and guidelines, EN should be started within 24–48 h in SAP or pSAP in order to prevent gut barrier dysfunction and dysmotility and infectious complications. EEN is also associated with a shorter duration of hospitalization and lower hospital costs [46,47].

**Table 3 nutrients-13-01498-t003:** Summary of articles regarding comparison of early and delayed enteral nutrition in SAP.

Author	Findings	Type of Analysis	Outcomes
Sun et al. [48]	Lower CD4+ T-lymphocyte %, CD4 +/CD8+ ratio, CRP Higher IgG and HLA-DR in EEN Lower SIRS, MODS, and pancreatic infection rates Lower duration of hospitalization in the ICU in EEN	Randomized controlled trial including 60 patients Comparison of EEN (48 h) and DEN (8th day) in SAP	EEN improves the course, but not decreases mortality compared to DEN in SAP patients
Sun et al. [49]	EEN does not increase IAP Decreased AP severity and clinical course, but did not decreased mortality in EEN	Randomized controlled trial including 60 patients Comparison of EEN (48 h) and DEN (8th day): impact on IAP and disease severity in SAP	EEN improves the course, but does not decrease mortality compared to DEN, EEN does not increase IAP in SAP patients
Zou et al. [50]	Lower hospital mortality, duration of hospitalization, % of patients requiring mechanical ventilation, surgery, continuous renal replacement therapy Lower incidence of local and systemic septic complications, acute kidney injury EEN	Retrospective analysis of 93 patients Comparison of EEN (within 72 h) and DEN (later than 72 h, within 7 days) in SAP	EEN should be started within 72 h of SAP onset
Vaughn et al. [51]		Systematic review including 11 RCTs (11 RCTs on SAP) (948 patients) Comparison of EEN (≤48 h) and DEN (>48 h) in all severity degrees of AP	No difference in outcomes between EEN and DEN in SAP patients
Bakker et al. [52]	Lower rate of complications in EEN	Meta-analysis of 8 RCTs (165 patients) Comparison of EEN (≤24 h) and DEN (>24 h) in all severity degrees of AP	EEN is associated with a reduction of complications
Bakker et al. [53]	Comparable rates of complications and mortality	Multicenter RCT including 208 patients Comparison of EEN EEN with an oral diet at 72 h of admission in SAP	EEN is not superior to an oral diet after 77 h in SAP patients
Wereszczyńska-Siemiątkowska et al. [54]	Lower mortality rate, frequency of infected necrosis/fluid collections, respiratory failure, and a need for ICU hospitalization in EEN	Retrospective analysis of 197 patients Comparison of EEN (≤48 h) and DEN (>48 h) in pSAP	EE in SAP should be started within 48 h after admission to hospital
Song et al. [55]	Lower mortality, MOF, surgery, systemic and local infection rates in EEN Comparable SIRS and other local complication rates in EEN	Meta-analysis including 10 RCTs (1051 patients) Comparison of EEN (≤48 h) and DEN (>48 h) or PN in pSAP, SAP	EEN is efficient and safe in pSAP and SAP patients
Li et al. [56]	Lower rate of overall infectious, catheter-related septic and local infectious complications lower hyperglycemia, shorter length of hospital stay, decreased mortality in EEN Comparable pulmonary complications	Meta-analysis of 11 studies (775 patients) Comparison of EEN (≤48 h) and DEN (>48 h) in pSAP	EEN improves the outcome and reduces complication rate in pSAP and SAP patients
Qi et al. [57]	Lower number of local infectious complications and MODS only in EEN in pSAP and SAP	Meta-analysis including 8 studies (727 patients) Comparison of EEN (<24 h) with DEN, PN in with all AP severity degrees	EEN should be used only in pSAP and SAP patients (not lower degrees) No advantages of EEN in MAP and MSAP patients

SAP, severe acute pancreatitis; pSAP, predicted severe acute pancreatitis; EEN, early enteral nutrition; DEN, delayed enteral nutrition; PN, parenteral nutrition; RCT, randomized controlled trial; MODS, multiorgan dysfunction syndrome; MOF, multiorgan failure; ICU, intensive care unit, IAP, intra-abdominal pressure.

In our opinion, in pSAP and SAP patients, EEN should be started within 48 h of admission to hospital. Starting EN within the first 24 h is frequently not possible because of clinical status (abdominal pain, nausea, vomiting) and metabolic disturbances (acid-base balance, dehydration, electrolyte deficiency), which should be first controlled. Besides, during first 24 h, diagnostic process is performed to assess the severity of AP. In accordance with above mentioned information, prediction of SAP using some scores (the Ranson, Glasgow scores) requires 48 h, and for patients without pSAP, EN is not recommended.

## 7. The Nasogastric versus Nasojejunal Tube in Enteral Nutrition of SAP Patients

Based on above mentioned reports, it is obvious that EEN is superior to PN in AP patients. But what about the enteral rout? Which rout is optimal for SAP patients. Historically, the nasojejunal tube (NJT) was preferred in EN in order to minimize pancreatic secretion. According to the current mentioned above theory of “no pancreatic rest”, the nasogastric tube (NGT) might be preferred in SAP patients, because it has been proven that insertion of the feeding tube in the stomach does not increase pancreatic secretion in SAP patients, and theoretically gastric placement of the feeding tube is easier. Thus, an opinion on the optimal rout of EN has been also changed. For 50 years, EE using NJT feeding was considered contraindicated in AP. According to “pancreatic rest” theory, TPN was used in all SAP patients. Similarly due to this theory, NGT was primarily contraindicated in SAP patients. There are numerous studies comparing NGT and NJT in the literature. Summary of articles regarding comparison of NGT and NJT in SAP is presented in Table 4.

Based on the mentioned above studies, generally EN may be provided by the NGT and NJT in SAP patients. NJT seems to be better than NGT in cases of gastroparesis associated with an aspiration risk, swelling of the pancreas, or large post inflammatory pancreatic cysts impressing the stomach or duodenum. However, insertion of the NJT is more complicated, more difficult and frequently it must be performed with an endoscopic approach or under fluoroscopic guidance, and may need additional sutures or clips in order to fix its placement [18]. It should be added that longer duration of NGT or NTJ can lead to complications such as discomfort for a patient, dislocation or unintentional tube removal, aspiration, sinusitis, and injury of the nasal cavity. Therefore, for patients requiring enteral feeding for a long time (>30 days), according to general nutritional recommendations, percutaneous gastrostomy or microjejunostomy should be considered [18].

We prefer enteral feeding via NGT in most patients, and NJT in patients with gastric outlet obstruction, because the insertion of NGT is easier and does not require endoscopic control in contrast to NJT.

**Table 4 nutrients-13-01498-t004:** Summary of articles regarding comparison of nasogastric versus nasojejunal tube in enteral nutrition in SAP.

Author	Findings	Type of Analysis	Outcomes
Eatock et al. [58]	Comparable outcome in NGT and NJT Mortality (18.5%) in NGT and (30.4%) in NJT patients	Pilot Randomized control trial including 50 patients Comparison of NGT and NJT in EE in SAP	EN via NGT was easier and equally effective compared to EN via NJT in SAP patients
Singh et al. [59]	Comparable rate of infectious complications, abdominal pain during refeeding, bowel permeability, and endotoxemia in both groups	Randomized control trial including 78 patients Comparison of NGT and NJT in EE in SAP	EE via NGT comparable to EE via NJT in SAP patients
Petrov et al. [60]	Comparable effects including mortality and feeding intolerance in both groups	Meta-analysis including 4 trials (92 patients) Comparison of NGT and NJT in EE in pSAP	EE via NGT safe and well tolerated in pSAP patients
Chang et al. [61]	Comparable mortality, and complications (tracheal aspiration, diarrhea, increased abdominal pain), covering of energy requirement in both groups	Meta-analysis including 3 trials (157 patients) Comparison of NGT and NJT in EE in pSAP	EE via NGT safe and well tolerated in pSAP patients
Nally et al. [62]	Comparable covering of the energy requirement, tolerance of enteral feeding, increase of abdominal pain and tube displacement was similar in both groups	Meta-analysis including 4 RCT Comparison of NGT and NJT in EE in SAP	NGT feeding is efficacious in 90% of SAP patients
Dutta et al. [63]	Comparable mortality, MODS, infectious complications, tube insertion and enteral feeding related complications, indications for surgery, intolerance of enteral feeding with necessity of PN administration, increased abdominal pain in both groups	Meta-analysis including 5 RCT (220 patients) Comparison of NGT and NJT in EE in SAP	Insufficient evidence regarding superiority/inferiority/equivalence between NGT and NJT in EE in SAP patients

SAP, severe acute pancreatitis; pSAP, predicted severe acute pancreatitis; EN, enteral nutrition; RCT, randomized controlled trial; MODS, multiorgan dysfunction syndrome.

## 8. Composition of Enteral Nutrition Formulas in SAP Patients

Historically, it has been thought that elemental and semi-elemental formulas less stimulate pancreatic secretion, are associated with a lower digestion, and are easily absorbed within a small bowel. Currently, the use of polymeric formulas can be sufficient and useful in SAP patients [19]. In a randomized prospective pilot study comparing a semi-elemental formula with a polymeric formula in enteral feeding in AP patients, tolerance of both formulas was good. In both groups, steatorrhea and creatorrhea were lower than normal. A significantly shorter duration of hospitalization and a lower loss of weight (*p* < 0.05) were noted in the patients receiving a semi-elemental formula. Therefore, this study showed comparable food tolerance in both groups, but better clinical outcomes in those patients receiving a semi-elemental formula [64].

A meta-analysis including 20 RCTs (1070 patients, involving 825 patients with SAP) showed no associations between enteral feeding intolerance and a kind of enteral formula (semi-elemental or polymeric formula), the probiotics administration, and immunomodulating nutrition. The infectious complications and mortality rates were similar in compared groups. Regarding formula composition, the authors concluded that administration of both polymeric and semi-elemental formulas was associated with a similar risk of feeding intolerance, infectious complications or mortality in AP patients [65].

A retrospective study including 948 patients (382 patients receiving the elemental formula and 566 patients in the control group) demonstrated a similar incidence of mortality (10.2% vs. 11.0%), sepsis (5.0% vs. 7.1%), hospital-free duration (54 days vs. 51 days), and total health-care costs ($29,360 vs. $34,214) in the two groups. Thus, this large study showed comparable results of enteral feeding with the use elemental, semi-elemental and polymeric formulas in AP patients. This study involved 817 patients with SAP [66].

Reviewing the literature, we found a study on the association between a polymeric formula in EEN and chylous ascites (CA) in SAP patients. Zhang et al. [67] described CA in SAP patients receiving EEN with a polymeric formula. This retrospective study included SAP 85 patients. The SAP patients were divided into two groups according to timing of EN introduction: EEN (<72 h) and DEN (>72 h). The chylous ascites was noted in 13 (15.29%) of 85 patients. CA was more frequently reported in patients receiving EEN patients with the use of a polymeric formula (9/33, *p* < 0.05). Duration of hospitalization in the ICU and in mortality rate were comparable regardless the CA presence. The study demonstrated a higher CA incidence in patients receiving EEN with the use of a polymeric formula in SAP patients, but further studies are needed to confirm these observations [67].

According to most authors, polymeric and elemental formulas are comparable regarding the nutrition tolerance and impact on clinical outcome in SAP patients. Summary of articles regarding composition of enteral formulas in SAP is presented in Table 5.

## 9. Immunomodulating Nutrition (IN) in SAP Patients

It is known that in SAP patients, systemic immune response is deteriorated due to the cytokine storm. Therefore, theoretically, modulation of the altered immune response would be indicated in SAP patients. Immunomodulating nutrition (immunonutrition, IN) involves four main immunonutrients as follows: glutamine (Gln), arginine (Arg), omega-3-unsatturated fat acids (PUFA), and nucleotides [68]. IN is widely used in surgical and oncological patients and its role in these patient groups has been well documented [50,51]. The use of IN in SAP patients is controversial.

### 9.1. Immunonutrients

Gln is a conditionally essential amino acid. It is a preferred fuel for enterocytes, lymphocytes and neutrophils. It also important fort the function of gut-associated lymphoid tissue (GALT) and respiratory immunity. It is a substrate in the synthesis of glutathione as well as increases expression of heat shock protein (HSP). Therefore, it is essential for the modulation of altered immune response and gut barrier dysfunction in SAP patients [68]. Arg, the second immunonutrient, is a semi-essential amino acid which plays an important role in protein synthesis. It stimulates activity of T lymphocytes and phagocytosis of neutrophils [68]. PUFA also modulate the immune response. Omega-3 fatty acids decrease the hyper-inflammatory response and immunosuppression [68]. Nucleotides are important for the proliferation of immune cells and they are also necessary in wound healing. They influence on T lymphocytes function [68,69].

There are no advantages of EN supplemented with immunonutrients [70,71,72,73,74,75], but parenteral administration of Gln and PUFA has been associated with a better outcome compared to the control PN with no immunonutrients [76,77,78,79,80,81,82,83]. 

So, the role of IN in the nutritional support for SAP is questionable and needs further investigations. Di Martino et al. [84] are going to assess the advantages and disadvantages of different types of nutritional supplementation such as Gln, Arg, PUFA, and nucleotides in EN and PN in AP patients. A research protocol of a planned large meta-analysis was described in Cochrane Database Systematical Review in 2019 [84].

### 9.2. Probiotics

It is known the deterioration of the gut barrier function, intestinal microbiota, and bowel motility in AP increase risk of bacterial translocation and risk of septic complications. Therefore, numerous studies have been performed in order to assess the role of probiotics for improving the gust function, and some authors have shown the advantages, but these findings are inconsistent and need further investigations. Currently ESPEN does not recommend the use of probiotics in the nutritional support in SAP patients [27]. Summary of some trials regarding IN and other supplements in SAP is listed in Table 6.

We recommend using Gln supplementation in TPN to prevent damage to enterocytes and intestinal barrier dysfunction in SAP patients feeding parenterally. In our opinion, PUFA are also useful in PN for modification of disturbed immune response in SAP patients. We do not recommend the use of probiotics because of not sufficient data of their benefits in SAP patients.

**Table 6 nutrients-13-01498-t006:** Summary of articles regarding immunonutrition and other supplements in SAP.

Author	Findings	Type of Analysis	Outcomes
**Enteral Immunonutrition**			
Petrov et al. [70]	Comparable risk of infectious complications and mortality, duration of hospitalization in both groups	A meta-analysis including 3 RCTs (78 patients) Comparison of IN and standard enteral formula in AP patients (from MAP to SAP)	No benefits of IN in EE in AP patients (including SAP patients
Poropat et al. [71]	Comparable overall mortality and SIRS rate in both groups	A meta-analysis including 3 RCTs (78 patients) Comparison of IN and standard enteral formula in AP patients (from MAP to SAP)	No benefits of IN in AP patients
Pearce et al. [72]	Comparable decreased CRP in both groups	Randomized controlled trial including 31 patients Comparison of EIN and control feeding in pSAP patients	The cause of the unexpectedly higher CRP values in the study group is unclear
Huang et al. [73]	Comparable APACHE II score, duration of hospitalization, costs in both groups	Randomized controlled trial including 32 patients Comparison of EIN and control feeding in pSAP patients	EIN (Gln, Arg) improves the gut barrier function by reducing the gastrointestinal permeability and decreasing plasma endotoxin level in the early SAP phase
Singh et al. [74]	Comparable infectious complications, prealbumin level, total duration of hospitalization/duration of hospitalization in ICU, and mortality in both groups	Randomized controlled trial including 80 patients Comparison of EIN (Gln) and control feeding in pSAP patients	No significant impact of Gln on gut permeability in SAP patients
Arutla et al. [75]	Comparable rated of infected necrosis and in-hospital mortality in both groups Higher increase of serum Gln, lower polyethylene glycol, higher decrease of Il-6 in Gln group	Randomized controlled trial including 40 patients Comparison of standard nutrition and standard nutrition supplemented with enteral Gln in SAP and pSAP patients	Enteral Gln supplementation improves the gut permeability and oxidative stress in SAP and pSAP patients
**Parenteral immunonutrition**			
Jafari et al. [76]	Lower mortality rate, shorter duration of hospitalization PIN group	Meta-analysis including 7 RCTS on PIN supplemented with Gln and/or PUFA	PIN (Gln, PUFA) can improve prognoses in patients with AP
Fuentes-Orozco et al. [81]	Increased IL-10, total lymphocyte and lymphocyte subpopulation counts, and albumin levels, improvement of nitrogen balance, lower rate of infectious complications in Gln group Comparable duration of hospitalization and mortality rate in both groups	Randomized controlled trial including 44 patients Comparison of standard PN (n = 22) and Gln-supplemented PN in SAP patients	PIN (Gln) may decrease infectious morbidity rate
Xu et al. [82]	Shorter duration of acute respiratory distress syndrome, renal insufficiency, acute hepatitis, shock, encephalopathy, and paralytic ileus, and hospitalization, lower APACHE II score, lower infection, surgery and mortality rates in early group	Randomized controlled trial including 80 patients Comparison of 2 groups of intravenous Gln (early treatment group) or 5 d after (late treatment group) admission in SAP patients	Early Gln supplementation superior to delayed in SAP patients
Wang et al. [83]	Higher eicosatetraenoic acid (EPA), lower CRP level, better oxygenation index, shorter duration of continuous renal replacement therapy in PUFA group	Randomized controlled trial including 40 patients Comparison of standard PN and PN supplemented with omega-3-fatty acids	PN supplemented with PUFA diminished the hyperinflammatory response by the EPA increase and the proinflammatory cytokine decrease in SAP patients
**Probiotics**			
Gou et al. [85]	No impact of probiotics on pancreatic infection, total infections, operation, mortality rates, duration of hospitalization	Meta-analysis including 6 trials (536 patients) Analysis of advantages and disadvantages of probiotics on the outcome in pSAP patients	No sufficient data to draw conclusions on the role of probiotics in nutrition in pSAP patients
Besselink et al. [86]	Higher infectious complications, mortality, bowel ischemia rates in probiotics group	Multicenter randomized controlled trial including 298 pSAP patients Comparison of probiotic sand placebo groups	Probiotics do not decrease a risk of septic complications in pSAP patients Use the probiotic prophylaxis is not recommended in SAP patients
Wang et al. [87]	Lowest pancreatic infectious complications, MODS, mortality rate, TNF-α and IL-6 levels, highest Il-10 as well as APACHE II scores in EN + EcoIN	Randomized controlled trial including 183 SAP patients Comparison of receiving PN, EN, or EN + EcoIN	Combination of EcoIN with EN has got more advantages compared to exclusive EN in SAP patients

SAP, severe acute pancreatitis; pSAP, predicted severe acute pancreatitis; EN, enteral nutrition; EIN, enteral immunonutrition; PN, parenteral nutrition; PIN, parenteral immunonutrition; RCT, randomized controlled trial; MODS, multiorgan dysfunction syndrome; ICU, intensive care unit.

## 10. Antisecretory Management

Some authors recommend the use of somatostatin, a hormone suppressing the pancreatic secretion in SAP patients. It is administered to rest by inhibiting pancreatic secretions and to prevent autodigestion stimulated by enteral nutrition. In practice, octreotide is used because of its longer half-life compared with somatostatin (72–98 vs. 2–3 min) and simpler administration (intravenous continuous infusion vs subcutaneous three doses per day) [20].

The data regarding the use of somatostatin/octreotide are not sufficient in order to recommend their standard use in SAP. We recommend intravenous administration of inhibitors of protein pomp due to the higher risk of stress ulcer in SAP.

## 11. Summary

In summary, it is certain and indisputable that nutritional support is necessary in all SAP patients in order to improve the nutritional status as well as to modulate an altered immune response, prevent the gut barrier dysfunction, bacterial translocation and infectious complications. The early nutritional intervention is recommended. The theories of the “pancreatic rest” and “bowel rest” are not current. Therefore, the route for the nutritional support in SAP patients should be as physiological as possible. Oral nutrition is recommended for patients with MAP and MSAP. The early (started within 24–48 h of admission) enteral nutrition is optimal for SAP patients who do not tolerate oral nutrition due to the disease severity. Currently, both NGT and NJT can be used for EN in SAP patients. NJT would be better than NGT in patients with gastroparesis associated with an aspiration risk, swelling of the pancreas, or large pancreatic post-inflammatory cysts impressing the stomach or duodenum. When prolonged EN is planned, insertion of percutaneous gastrostomy or jejunostomy would be considered. IN has a limited role in SAP patients. There are no benefits of enteral IN and currently it is not recommended. In SAP patients, in whom EN is contraindicated, PN should be administered. It is important that PN is recommended only in patients with impossibility of EN. It has been proven that parenteral IN with Gln and PUFA supplementation is beneficial for SAP patients. Therefore, it is recommended in them. There are contradictory reports regarding the advantages and disadvantages of the probiotic prophylaxis in SAP patients. Currently, they are not recommended in SAP patients.

In conclusion, currently, all SAP patients, as the patients of high nutritional risk, should be supported by the early nutrition which should be started as soon as possible. All our recommendations based on this review are summarized in Table 7. Algorithm of nutritional management in SAP is presented in Figure 1.

## 12. The Other Clinical Considerations and Practical Tips Regarding Nutritional Support in Patients with SAP

Nutritional support should be individually composed. The nutritional risk is assessed according to commonly used nutritional risk score such as Subjective Global Assessment (SGA) and Nutritional Risk Score 2002 (NRS 2002) [27]. The energy requirement should be estimated using indirect calorimetry (IC) if possible, or should be calculated by the formula 25–35 kcal/kg/d. The formulas of enteral and parenteral nutrition are composed based on the following nutritional requirements: protein 1.2–1.5 g/kg/d, carbohydrates 3–6 g/kg/d corresponding to blood glucose concentration (aim: <10 mmol/L), lipids up to 2 g/kg/day corresponding to blood triglyceride concentration (aim: <12 mmol/L), Natrium 1–2 mmol/kg/d, potassium 1–2 mmol/kg/d, chlorine 2–4 mmol/kg/d, phosphorus 0.1–0.5 mmol/kg/d, magnesium 0.1–0.2 mmol/kg/d, and calcium 0.1 mmol/kg/d. They should be modified depending on serum concentrations, metabolic status and balance [88].

Formulations used in EE and PN should contain proteins, carbohydrates and fats. In complete PN, solutions of vitamins and micronutrients should be administered. EE via NGT is administered in interrupted boluses (200–300 mL 5–6 times per day under control of gastric residual volume (GRV)) or continuous infusion (30–50 mL/h), while EE via NJT is administered by continuous infusions. The flow velocity should increase gradually: from 20–30 mL/h to 100–125 mL/h. In order to avoid complications (regurgitation, aspiration, or pneumonia), EN via NGT should be interrupted in GRV > 200 mL. EE should cover minimally 60% of energy requirement. When it is not possible or in intolerance of EE, PN should be added. PN should be started in volume of 50% of estimated nutritional requirement on day 1, 75%—on day 2, and 100%—from day 3. The hemodynamic status should be controlled and all disturbances/deficiencies of water/electrolytes and acid-base balance should be compensated for before starting nutrition to avoid re-feeding syndrome. In cases of EE intolerance such as diarrhea, the velocity of feeding should be decreased. When it is not sufficient, PN administration of PN should be considered. The assessment of nutritional requirement and control laboratory investigations should be performed minimally once a week for optimal nutritional support and modification of the type or formula if it is indicated. Also, a care of the tube (in EE) or catheter (in PN) is very important to avoid infectious and other catheter and tube related complications [89].

## Figures and Tables

**Figure 1 nutrients-13-01498-f001:**
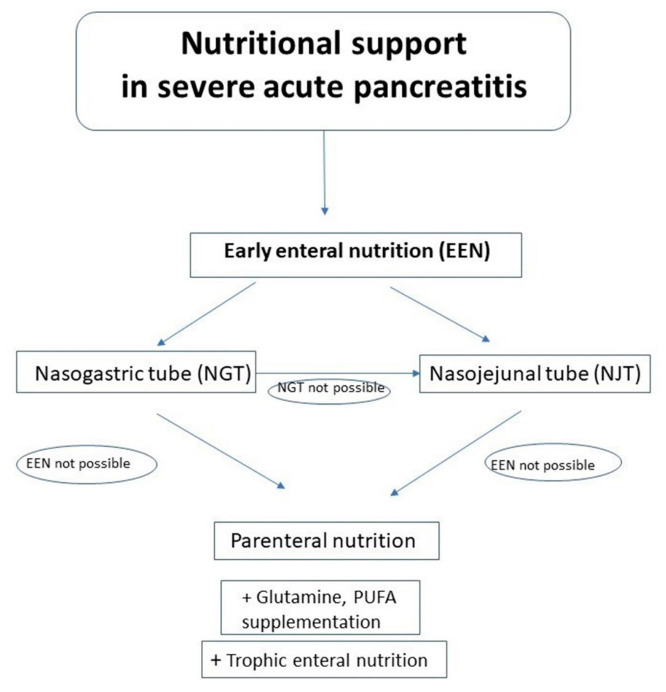
Algorithm of nutritional management in SAP.

**Table 1 nutrients-13-01498-t001:** Severity grading of acute pancreatitis according to the revised Atlanta classification [4].

Severity of Acute Pancreatitis	Description
Mild acute pancreatitis	No organ failure No local/systemic complications
Moderate severe acute pancreatitis	Organ failure < 48 h
Severe acute pancreatitis	Organ failure > 48 h Local/systemic complications

**Table 2 nutrients-13-01498-t002:** Summary of articles regarding comparison of enteral and parenteral nutrition in SAP.

Author	Findings	Type of Analysis	Outcomes
Cao et al. [39]	Lower rate of infectious complications, local complications, organ failure, MODS and mortality in EN Complication rates comparable	Meta-analysis including 6 RCTs (224 patients) Comparison of EN and PN in SAP	EN safer compared to PN in SAP patients
Al-Omran et al. [33]	Lower rate of mortality, MODS, septic complications, and indications for surgery than in EN	Meta-analysis including 8 (5 regarding SAP) RCTs (348 patients) Comparison of EN and PN in AP	EN should be standard nutritional intervention in AP
Yi et al. [34]	Lower mortality, infections, MODS, and surgery rates in EE. Comparable duration of hospitalization and nutrition	Meta-analysis including 8 RCTs (381 patients) Comparison of EN and PN in SAP	EE superior to TPN in SAP patients
Li et al. [40]	Lower rate of mortality, complications, MODS and surgery, shorter duration of hospitalization in EN	Meta-analysis including 9 RCTs (500 patients) Comparison of EN and PN in SAP	EE is preferred rather than TPN in SAP patients
Wu et al. [41]	Lower rate of mortality and infectious complications, shorter duration of hospitalization in EN. Comparable MODS rate in EE and PN.	Meta-analysis including 11 RCTs (562 patients) (348 patients) Comparison of EN and PN in SAP	EN recommended as an initial treatment for patients with SAP
Yao et al. [35]	Lower rate of mortality and MODS in EE.	Meta-analysis including 5 RCTs (348 patients) Comparison of EN and PN in SAP	EN should be recommended as the preferred route of nutrition for critically ill patients with SAP
Tao et al. [42]	Lower rate of infectious complications, MODS and mortality, shorter total duration of hospitalization and duration of hospitalization in the ICU.	Retrospective analysis of 185 patients Comparison of EE and PN in SAP	EE superior to PN in SAP patients
Gupta et al. [43]	Lower rate of respiratory and non-respiratory organ failure, shorter duration of hospitalization, lower cost of hospitalization in EE	Randomized controlled trial (17 patients) Comparison of EE and PN in SAP	EE safer and less expensive than PN in SAP patients

SAP, severe acute pancreatitis; EN, enteral nutrition; PN, parenteral nutrition; RCT, randomized controlled trial; MODS, multiorgan dysfunction syndrome; ICU, intensive care unit.

**Table 5 nutrients-13-01498-t005:** Summary of articles regarding composition of enteral formulas in SAP.

Author	Findings	Type of Analysis	Outcomes
Tiengou et al. [64]	Comparable feeding tolerance in both groups Shorter duration of hospitalization, lower loss of weight in patients receiving a semi-elemental formula.	Randomized controlled trials including 30 patients Comparison semi-elemental and polymeric formula in AP patients stratified according to severity	Comparable food tolerance in both groups, but better clinical outcome in patients receiving a semi-elemental formula
Petrov et al. [65]	Comparable feeding tolerance, infectious complications and mortality rates in both groups	Meta-analysis including trials (1070 patients) Comparison of semi-elemental or polymeric formula Comparison of semi-elemental and polymeric formula	The use of polymeric formula, compared to semi-elemental formula, does not lead to increased feeding intolerance, infectious complications or mortality
Endo et al. [66]	Comparable mortality, sepsis rates, hospital-free duration, total health-care costs in both groups.	Retrospective cohort study including 382 patients Comparison of elemental or control formula	Comparable results of EN with the use elemental, semi-elemental and polymeric formulas

SAP, severe acute pancreatitis; pSAP, predicted severe acute pancreatitis; EN, enteral nutrition; RCT, randomized controlled trial; MODS, multiorgan dysfunction syndrome.

**Table 7 nutrients-13-01498-t007:** Summary of our recommendations on nutritional support in SAP based on the literature review.

Aspect of Nutritional Support in SAP	Our Recommendations
The optimal route of feeding	EN is feeding of choice in SAP patients in whom oral nutrition is impossible Parenteral nutrition is reserved for patients with intolerance or impossibility of EE
The optimal timing of nutrition	EEN (<48 h of admission) is superior to DEN EN should be started within 48 h of admission
NGT versus NJT	NGT is the route of choice NJT is preferred in patients with GOO
Immunonutrition	IN supplementation (including Gln in dose 0.3–0.5 g/kg/d) is recommended in PN
Probiotics	Not recommended

SAP, severe acute pancreatitis; EE, enteral nutrition; EEN, early enteral nutrition; DEN, delayed enteral nutrition; PN, parenteral nutrition; NGT, nasogastric tube; NJT, nasojejunal tube; GOO, gastric outlet obstruction; IN, immunonutrition; Gln, glutamine.

## Data Availability

Not applicable.

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
