# Peer review of "Nutritional Support in Patients with Severe Acute Pancreatitis-Current Standards"

_nutrients, 2021, doi:10.3390/nu13051498_

Round 1

Reviewer 1 Report

This review describe the role of the nutritional support in patients with severe acute pancreatitis. The rational behind the study was clear and straight forward. The manuscript is almost well written.

While many different sources are used to set up the study in the introduction, little previous evidence is stated. The introduction is thus short and poorly sets up the rationale for the study. More attention to how this study fits into previous work in pancreatitis and inflammation should be added to improve this section. Please refer to doi: 10.3390/antiox9090781, 10.3390/antiox9100992. 

There are some minor grammar issues that should be fixed in order to aid the accessibility of the discussion to the reader.

Author Response

Dear Editor,

Dear Reviewer,

Thank you for peer reviewing of our manuscript  nutrients-1156590, entitled " Nutritional Support in Patients with Severe Acute Pancreatitis - Current Standards".

Thank you for your questions and comments. We have fully addressed all the comments and my responses appear below. Our revised work includes corrections according to reviewers’ comments in the text. The changes, made according to reviewers’ comments, are highlighted in red print in the text.

We take this opportunity to express my gratitude to the reviewers for their constructive and useful remarks. Their comments allowed us to identify areas in my manuscript that needed modification.

We also thank you for allowing me to resubmit a revised copy of the manuscript.

We hope that the revised manuscript is now acceptable for publication in Nutrients.

Yours sincerely,

Beata Jabłońska.

Responses to Reviewer 1.

  1. The introduction is thus short and poorly sets up the rationale for the study. More attention to how this study fits into previous work in pancreatitis and inflammation should be added to improve this section. Please refer to doi: 10.3390/antiox9090781, 3390/antiox9100992. 

Answer: The introduction has been extended. The proposed citations  [5,6] (5. Fusco R, Cordaro M, Siracusa R, D’Amico R, Genovese T, Gugliandolo E, Peritore AF, Crupi R, Impellizzeri D, Cuzzocrea S, Di Paola R. Biochemical Evaluation of the Antioxidant Effects of Hydroxytyrosol on Pancreatitis-Associated Gut Injury. Antioxidants. 2020; 9(9):781. https://doi.org/10.3390/antiox9090781. 6. Cordaro M, Fusco R, D’Amico R, Siracusa R, Peritore AF, Gugliandolo E, Genovese T, Crupi R, Mandalari G, Cuzzocrea S, Di Paola R, Impellizzeri D. Cashew (Anacardium occidentale L.) Nuts Modulate the Nrf2 and NLRP3 Pathways in Pancreas and Lung after Induction of Acute Pancreatitis by Cerulein. Antioxidants. 2020; 9(10):992. https://doi.org/10.3390/antiox9100992).

have been added as follows:

  1. Introduction
    • Definition and epidemiology of AP

Acute pancreatitis (AP) is an inflammatory disease involving a pancreatic parenchyma and peripancreatic tissues with a potential systematic immune response in a severe disorder course. Recently, the incidence of AP has been increased in most countries. The mean global AP incidence is 34/100 000 [1]. The incidence of AP in Europe ranges from 4.6 to 100/100 000 [2]. Poland belongs to the countries with the highest incidence rate – 72.1/100 000 [3].

  • Classification of AP

The revised Atlanta classification (2012) [4] distinguished interstitial edematous (1) and necrotizing (2) acute pancreatitis types. In most patients, the first AP type is observed, but in 5-10% of cases, the necrotizing AP occurs. According to the updated Atlanta classiffication, necrotizing AP most frequently involves the whole pancreas and peripancreatic tissues, less frequently only the peripancreatic tissues, and rarely only pancreas without adjacent tissues [4]. The authors of the Atlanta classiffication distinguished two overlapping AP phases: early and late. The early phase usually lasts for the first week, and a second delayed phase can be prolonged to weeks or months [4]. In the early phase, the local pancreatic inflammation leads to generalized immune alterations as a response for the pancreatic injury. The pancreatic inflammatory injury triggers the cytokine storm which is demonstrated as the systemic inflammatory response syndrome (SIRS). The prolonged SIRS can lead to the multiorgan dysfuntion syndrome (MODS) which determines the AP severity. In the second delayed phase, systemic and/or local complications are observed. This phase is noted only in patients with moderately severe or severe acute pancreatitis [4].

  • Pathomechanism of AP, the role of oxidative stress in AP

In AP, the secretion of digestive enzymes is disturbed. Digestive enzymes are transported outside the cells. Additionally, in damaged pancreas, the separation of digestive enzymes from lysosomal hydrolases is disturbed. Therefore, both types of enzymes are colocalized within intracellular vacuoles. This colocalization phenomenon causes the premature activation of digestive enzymes. Subsequent rupture of these vacuoles liberates the activated digestive enzymes in the cytoplasmic space, followed by a cascade of events leading to AP. The activation of pancreatic enzymes in and around the pancreas and bloodstream leads to the pancreatic  coagulation necrosis as well as necrosis and hemorrhage of peripancreatic and peritoneal adipose tissue. Oxydative stress plays an important role in the early phase of AP. Moreover, in AP,  release of proinflammatory cytokines and mediators and the increased intestinal permeability and is noted. Intestinal barrier dysfunction results in infected necrosis, bacteremia and multiorgan  failure. Therefore, supporting of the alimentary tract is very important in the AP treatment. Oxidative stress increases the inflammatory infiltration of within the intestine, increased damage to intestinal barrier,  and allows bacterial translocation into the bloodstream. Therefore, antioxidants should be helpful in the AP treatment [5].  Some antioxidants (such as resveratrol, selenium, ascorbic acid, melatonin, hydroxythyrozol, cerulein), have been investigated as potential beneficial agents in AP patients, but these investigations have regarded mainly animal models, not human participants [6].

  • Assessment of the severity of AP

Management in AP depends on the severity of the disease. Therefore, identification of patients with potentially severe acute pancreatitis (SAP), who need nutritional support (NS), is very essential. The authors of the Atlanta classiffication defined and stratified the severity of AP. The severity grading of AP according to the Atlanta classification is presented in Table 1. Three types of AP severity have been identified: mild acute pancreatitis (MAP), moderately severe acute pancreatitis (MSAP), and severe acute pancreatitis (SAP). The presence of organ deficiency and local or systemic complications differs the above mentioned AP types. The organ failure it is not reported in MAP, it is transient in MSAP, and it is persistent (> 48 hours) in SAP. It can include single organ failure or multiple organ failure.. These patients require aggressive treatment including nutritional intervention [4].

Table 1. Severity grading of acute pancreatitis according to the revised Atlanta classification [4].

Severity of acute panreatitis

Description

Mild acute pancreatitis

No organ failure

No local/systemic complications

Moderate severe acute pancreatitis

Organ failure < 48 hours

Severe acute pancreatitis

Organ failure > 48 hours

Local/systemic complications

Assessment of the AP severity is crucial for the appropriate management. Several single parameters or more complex scores have been developed to assess the AP severity. The Ranson score is the oldest score for prediction of AP severity. It involves 11 parameters including 5 parameters assessed at admission (scores 0-5), and 6 parameters - 48 hours later (scores 0-6). Although a score ≥ 3 has a high sensitivity and specificity regarding a severe course of AP (83.9% and 78.0%, respectively) and a negative predictive value of 94.5%, the severity can be predicted no earlier than 48 hours after admission. The later introducted scores for prediction of the AP severity are as follows: the Glasgow (Imrie) score (8 parameters), Multiple Organ Dysfunction Score (MOSS/MODS) (12 parameters), Bedside Index of Severity of Acute Pancreatitis (BISAP) score (5 parameters), and the Acute Physiology and Chronic Health Evaluation (APACHE II) score (14 parameters). All the above mentioned scores include various clinical and laboratory parameters. The sensitivity and specificity of these scores for predicting SAP is 55- 90%. The inability to obtain a complete score until at least 48 hours of the disease is a limitation of the Ranson and Glasgow scores, while the complexity of the scoring system itself is a limitation of the APACHE II score (including age, Glascow coma score, vital and oxygenation parameters, as well as biochemical and hematological parameters). APACHE II score was originally developed to predict mortality in intensive care patients. The APACHE II score ≥ 8  predicts severe acute pancreatitis  (sensitivity of 65%-83%, and specificity of 77%-91%) [7].
BISAP score (including blood urea nitrogen (BUN), impaired mental status, the systemic inflammatory response syndrome (SIRS), age and pleural effusion) allows to predict the AP severity within the first 24 hours of hospitalization [7].

C-reactive protein (CRP) is the most useful single laboratory parameter. Despite its delayed increase (peaking not earlier than 72 hours after the onset of AP) it is accurate and widely available. An elevated CRP level > 150 mg/l is a prognostic factor for complicated course (sensitivity of 85%) in the first 72 hours after the onset of symptoms
[8,9]. The other biochemical parameters (albumin, creatinine, procalcitonin) can be useful in prediction of SAP. While their predictive value is unsatisfactory when used alone, their use, in combination, can effectively improve the sensitivity of prediction. Therefore, the CRP/albumin ratio and creatinine/albumin ratio have been reported to be useful in prediction of SAP [9].

Premature intraacinar activation of trypsinogen leads to the cytokine storm in AP. Therefore, various cytokines have been tested to be useful for prediction SAP. Among, them, proinflammatory interleukin-6 (Il-6) is the most useful parameters. It has been confirmed that the plasma level of IL‐6 is elevated early phase of AP. Therefore, it is a sensitive and specific marker to predict organ failure and SAP [10]. A few studies have shown that elevated levels of the tissue necrosing factor-α (TNF‐α) predicted SAP [11,12]. The opinions regarding the utility of the TNF‐α in prediction SAP are contradictory. According to another authors, it is elevated only in a small percentage (9-36%) of SAP patients, and it cannot be used as a prognostic factor in AP [10,13,14]. Similarly, the significance of IL‐10 regarding severity of AP is yet not clear because of the contradictory reports [10,11,15].The other cytokins that have been assessed for predicting AP are as follows: IL-8, and IL-18 [8].

Angiopoietin-2 (Ang-2), a type of glycoprotein acting selectively on endothelial cells and leading to an increase in endothelial permeability, has been identified to be a useful predictor for acute gastrointestinal injury and intestinal barrier dysfunction in patient with AP [16].

              The morphologic severity of AP can be determined using a CT severity index (CTSI) that was developed by Balthazar and then modified and extended to monitor organ failure by Silverman, Banks, and colleagues in 2004 [8].

Concluding, the following cutoff values positively correlate with the severity grade of AP: Ranson ≥ 3, Glasgow ≥ 3, MOSS ≥ 5, BISAP ≥ 2, APACHE II ≥ 8, CTSI ≥ 5, procalcitonin ≥ 0.5 ng/ml, CRP ≥ 150 mg/l, and IL-6 ≥ 50 pg/ml [8].

  • Disturbances of the nutritional status in SAP

SAP is a serious acute disease. In these patients, resting energy expenditure (REE) increases due to inflammation-induced hypermetabolism and/or septic complications. In SAP, protein catabolism and increased energy requirements are observed. The disease leads to undernutrition and disturbances of water / electrolyte and acid-base balance. Therefore, energy and protein requirements are higher in patients with SAP  [17,18,19]. The pancreatic damage and infectious complications are associated with hyperglycemia which must be controlled. Hyperglicemia is induced by damage to beta cells, insulin resistance, and infectious complications. Therefore, insulin therapy should be considered in nutritional support in patients with SAP. Acute pancreatitis is usually associated with deficiency of numerous vitamins and micronutrients such as vitamins B1, B2, B3, B12, C, A, folic acid, and zinc. Moreover, dysregulations in water / electrolytes and of acid‐base balance are noted in SAP. Hypocalcemia is observed in 40-60% of patients due to saponification of calcium. The other disturbances are as follows: hypomagnesemia, decreased parathyroid hormone release, and increased calcitonin levels, hypertriglicerydemia [19]. The level of triglicerydes must be closely monitored during parenteral nutrition including lipid emulsions. It has been noted that a great protein loss with a negative nitrogen balance occurred in 80% of patients with SAP [3]. The nitrogen loss can be up to 20–40 g/d. Negative nitrogen balance is associated with increased mortality [20].

Therefore, all patients with SAP are at nutritional risk and require nutritional intervention [17]. Moreover, just at the SAP beginning, damage to enterocytes due to microcirculatory injury and gut hypoperfusion leads to increased gastrointestinal permeability and translocation of systemic cytokines, toxins, and bacteria [18,20]. Therefore, the goal of the  nutritional support (NS) in AP is not only to prevent and treat malnutrition, but also modulate and decrease altered inflammatory response [21]. Nutrition constitutes the third step of AP management according to the „PANCREAS“ (perfusion, analgesia, nutrition, clinical and radiological assessment, endoscopy, antibiotics, surgery) acronym described by Khaliq et al [22]. This acronym is very useful in management with SAP patients [3,5]. In Khaliq’s et al [22] report, enteral nutrition within 48 hours (with or without the use of nasojejunal tube) led to reduction of mortality in SAP patients [22].

  • Comparison of management and nutritional support in acute and chronic pancreatitis

The overall management including nutritional support is generally different in patients with  AP and chronic pancreatitis (CP), but there are also some similarities. Regarding general management, in both AP and CP, conservative treatment plays the most essential role in most patients, and surgery is reserved for patients with complications of these diseases. Obviously, the complication types are different in both groups. The infected necrosis, abscess, ileus, gastrointestinal perforation, hemorrhage are typical indications for surgery in AP patients. In CP, indications for surgey are the following: duodenal or bile duct stenosis, pancreatic tumor, pancreatic lithiasis with a Wirsung duct dilatation, small duct pancreatitis with an intractable pain. Regarding nutritional support, in MAP, the early oral low-fat diet is recommended, while in SAP, enteral feeding is used. Parenteral nutrition is reserved only for patients with intolerance or impossibility of enteral nutrition. The majority of patients with CP can receive normal food with supplementation of exogenous pancreatic enzymes. Generally, 10%–15% of patients need oral nutrition supplementation and 5% require tube feeding. Oral elemental supplements are associated with a significant decrease pain and an improvement in nutrition indices. Additionally, fat-soluble (A,D,E,K) and water-soluble (vitamin B12, folic acid, thiamine) vitamins as well as minerals such as magnesium, iron, selenium and zinc should be administered in cases of confirmed deficiency. Parenteral nutrition is used in < 1% of patients with complication of CP such as duodenal stenosis, complex fistula, or malnourished patients with gastrointestinal dysfunction prior to surgery. Similar to nutrition in AP, oral and enteral routes of nutrition are preferable, and parenteral nutrition is reserved for patients in whom oral and enteral nutrition is not possible [23].

The role of supplementation of pancreatic enzymes is different in AP and CP patients. In patients with AP, generally these supplements are not recommened except obvious pancreatic exocrine insufficiency (PEI). The informations regarding occurence of PEI in AP patients during hospitalization are not sufficient. PEI can be caused by the pancreatic necrosis, but it is manifested following AP [24]. The knowledge of PEI after AP is better. PEI is noted at least during the first 6 to 18 months after AP, and the grade of dysfunction is related to the AP severity [25]. PEI is common in patients with CP. The low-fat oral diet in MAP, or special formula of enteral/parenteral nutrition in SAP (when oral diet is not possible) are commonly used in most patients. In CP, supplementation of exogenous pancreatic enzymes is very essential due to maldigestion and malabsorption in the disease course [26]. A large meta-analysis by Hollemans et al [27], containing 32 studies (1495 patients), showed the 27.1% prevalence of PEI after AP. It was positively correlated with degree of pancreatic parenchymal injury, alcoholic cause (22.7%), SAP (33.4%), and necrotizing pancreatitis (32%). Therefore, clinical observations in terms of symptoms of maldigestion (diarrhea, steatorrhea) and/or non-invasive pancreatic function investigations (such as fecal fat and fecal elastase) for at least 6-18 months after AP, especially in alcoholic, severe, and necrotizing pancreatitis are recommended. In clinically relevant PEI, supplementation of pancreatic enzymes is recommended [19,27].

Considering all mentioned above issues, it is obvious that the nutritional support is very essential to correct nutritional status and modify altered immune response in SAP patients. Some issues regarding the optimal timing, type (enteral nutrition (EN) vs parenteral nutrition (PN), via nasogastric tube (NGT) or nasojejunal tube (NJT)), and the role of immunonutrients, such as glutamine (Gln), Arginine (Arg), polyunsaturated omega-3-fatty acids (PUFA), nucleotids in nutrition of patients with SAP, are discussed in this review.

The aim of this article is to review the current literature on nutritional support in SAP patients as well as to discuss and resolve the above issues.

  1. There are some minor grammar issues that should be fixed in order to aid the accessibility of the discussion to the reader.

Answer: English has been improved and all grammar issues have been corrected in the whole manuscript.

Reviewer 2 Report

Please find my comments attached in the word document. 

Author Response

Dear Editor,

Dear Reviewer,

 Thank you for peer reviewing of our manuscript  nutrients-1156590, entitled " Nutritional Support in Patients with Severe Acute Pancreatitis - Current Standards".

Thank you for your questions and comments. We have fully addressed all the comments and my responses appear below. Our revised work includes corrections according to reviewers’ comments in the text. The changes, made according to reviewers’ comments, are highlighted in red print in the text.

We take this opportunity to express my gratitude to the reviewers for their constructive and useful remarks. Their comments allowed us to identify areas in my manuscript that needed modification.

We also thank you for allowing me to resubmit a revised copy of the manuscript.

We hope that the revised manuscript is now acceptable for publication in Nutrients.

Yours sincerely,

Beata Jabłońska.

Responses to Reviewer 2.

  1. Section 1 – i) what is the incidence of acute pancreatitis (AP) in Poland/Europe? How does it compare to the occurrence in Europe and the rest of the world? Statistics are missing in the introduction.

Answer: The incidence of AP in the world, Europe and Poland has been presented in the introduction as follows:

  • Definition and epidemiology of AP

Acute pancreatitis (AP) is an inflammatory disease involving a pancreatic parenchyma and peripancreatic tissues with a potential systematic immune response in a severe disorder course. Recently, the incidence of AP has been increased in most countries. The mean global AP incidence is 34/100 000 [1]. The incidence of AP in Europe ranges from 4.6 to 100/100 000 [2]. Poland belongs to the countries with the highest incidence rate – 72.1/100 000 [3].

  1. ii) A small paragraph to mention how AP/SAP differs in terms of overall management and/or nutrition management with recurring acute or chronic pancreatitis will present value added information (O’brien and Omer, Nutrition in Clinical Practice 2019; Narayanan et al. Nutrition in Clinical Practice 2020). A section on severity assessment for SAP is key- the authors seem to have missed this important point. Please include relevance to the APACHE II scores/serum C reactive protein levels – this is vital to assess severity.

Answer: Comparison of overall management and nutrition in acute and chronic pancreatitis, and assessment of severity of AP has been presented in the introduction as follows:

  • Comparison of management and nutritional support in acute and chronic pancreatitis

The overall management including nutritional support is generally different in patients with  AP and chronic pancreatitis (CP), but there are also some similarities. Regarding general management, in both AP and CP, conservative treatment plays the most essential role in most patients, and surgery is reserved for patients with complications of these diseases. Obviously, the complication types are different in both groups. The infected necrosis, abscess, ileus, gastrointestinal perforation, hemorrhage are typical indications for surgery in AP patients. In CP, indications for surgey are the following: duodenal or bile duct stenosis, pancreatic tumor, pancreatic lithiasis with a Wirsung duct dilatation, small duct pancreatitis with an intractable pain. Regarding nutritional support, in MAP, the early oral low-fat diet is recommended, while in SAP, enteral feeding is used. Parenteral nutrition is reserved only for patients with intolerance or impossibility of enteral nutrition. The majority of patients with CP can receive normal food with supplementation of exogenous pancreatic enzymes. Generally, 10%–15% of patients need oral nutrition supplementation and 5% require tube feeding. Oral elemental supplements are associated with a significant decrease pain and an improvement in nutrition indices. Additionally, fat-soluble (A,D,E,K) and water-soluble (vitamin B12, folic acid, thiamine) vitamins as well as minerals such as magnesium, iron, selenium and zinc should be administered in cases of confirmed deficiency. Parenteral nutrition is used in < 1% of patients with complication of CP such as duodenal stenosis, complex fistula, or malnourished patients with gastrointestinal dysfunction prior to surgery. Similar to nutrition in AP, oral and enteral routes of nutrition are preferable, and parenteral nutrition is reserved for patients in whom oral and enteral nutrition is not possible [23].

The role of supplementation of pancreatic enzymes is different in AP and CP patients. In patients with AP, generally these supplements are not recommened except obvious pancreatic exocrine insufficiency (PEI). The informations regarding occurence of PEI in AP patients during hospitalization are not sufficient. PEI can be caused by the pancreatic necrosis, but it is manifested following AP [24]. The knowledge of PEI after AP is better. PEI is noted at least during the first 6 to 18 months after AP, and the grade of dysfunction is related to the AP severity [25]. PEI is common in patients with CP. The low-fat oral diet in MAP, or special formula of enteral/parenteral nutrition in SAP (when oral diet is not possible) are commonly used in most patients. In CP, supplementation of exogenous pancreatic enzymes is very essential due to maldigestion and malabsorption in the disease course [26]. A large meta-analysis by Hollemans et al [27], containing 32 studies (1495 patients), showed the 27.1% prevalence of PEI after AP. It was positively correlated with degree of pancreatic parenchymal injury, alcoholic cause (22.7%), SAP (33.4%), and necrotizing pancreatitis (32%). Therefore, clinical observations in terms of symptoms of maldigestion (diarrhea, steatorrhea) and/or non-invasive pancreatic function investigations (such as fecal fat and fecal elastase) for at least 6-18 months after AP, especially in alcoholic, severe, and necrotizing pancreatitis are recommended. In clinically relevant PEI, supplementation of pancreatic enzymes is recommended [19,27].

  • Assessment of the severity of AP

Management in AP depends on the severity of the disease. Therefore, identification of patients with potentially severe acute pancreatitis (SAP), who need nutritional support (NS), is very essential. The authors of the Atlanta classiffication defined and stratified the severity of AP. The severity grading of AP according to the Atlanta classification is presented in Table 1. Three types of AP severity have been identified: mild acute pancreatitis (MAP), moderately severe acute pancreatitis (MSAP), and severe acute pancreatitis (SAP). The presence of organ deficiency and local or systemic complications differs the above mentioned AP types. The organ failure it is not reported in MAP, it is transient in MSAP, and it is persistent (> 48 hours) in SAP. It can include single organ failure or multiple organ failure.. These patients require aggressive treatment including nutritional intervention [4].

Table 1. Severity grading of acute pancreatitis according to the revised Atlanta classification [4].

Severity of acute panreatitis

Description

Mild acute pancreatitis

No organ failure

No local/systemic complications

Moderate severe acute pancreatitis

Organ failure < 48 hours

Severe acute pancreatitis

Organ failure > 48 hours

Local/systemic complications

Assessment of the AP severity is crucial for the appropriate management. Several single parameters or more complex scores have been developed to assess the AP severity. The Ranson score is the oldest score for prediction of AP severity. It involves 11 parameters including 5 parameters assessed at admission (scores 0-5), and 6 parameters - 48 hours later (scores 0-6). Although a score ≥ 3 has a high sensitivity and specificity regarding a severe course of AP (83.9% and 78.0%, respectively) and a negative predictive value of 94.5%, the severity can be predicted no earlier than 48 hours after admission. The later introduced scores for prediction of the AP severity are as follows: the Glasgow (Imrie) score (8 parameters), Multiple Organ Dysfunction Score (MOSS/MODS) (12 parameters), Bedside Index of Severity of Acute Pancreatitis (BISAP) score (5 parameters), and the Acute Physiology and Chronic Health Evaluation (APACHE II) score (14 parameters). All the above mentioned scores include various clinical and laboratory parameters. The sensitivity and specificity of these scores for predicting SAP is 55- 90%. The inability to obtain a complete score until at least 48 hours of the disease is a limitation of the Ranson and Glasgow scores, while the complexity of the scoring system itself is a limitation of the APACHE II score (including age, Glascow coma score, vital and oxygenation parameters, as well as biochemical and hematological parameters). APACHE II score was originally developed to predict mortality in intensive care patients. The APACHE II score ≥ 8  predicts severe acute pancreatitis  (sensitivity of 65%-83%, and specificity of 77%-91%) [7].
BISAP score (including blood urea nitrogen (BUN), impaired mental status, the systemic inflammatory response syndrome (SIRS), age and pleural effusion) allows to predict the AP severity within the first 24 hours of hospitalization [7].

         C-reactive protein (CRP) is the most useful single laboratory parameter. Despite its delayed increase (peaking not earlier than 72 hours after the onset of AP) it is accurate and widely available. An elevated CRP level > 150 mg/l is a prognostic factor for complicated course (sensitivity of 85%) in the first 72 hours after the onset of symptoms
[8,9]. The other biochemical parameters (albumin, creatinine, procalcitonin) can be useful in prediction of SAP. While their predictive value is unsatisfactory when used alone, their use, in combination, can effectively improve the sensitivity of prediction. Therefore, the CRP/albumin ratio and creatinine/albumin ratio have been reported to be useful in prediction of SAP [9].

      Premature intraacinar activation of trypsinogen leads to the cytokine storm in AP. Therefore, various cytokines have been tested to be useful for prediction SAP. Among, them, proinflammatory interleukin-6 (Il-6) is the most useful parameters. It has been confirmed that the plasma level of IL‐6 is elevated early phase of AP. Therefore, it is a sensitive and specific marker to predict organ failure and SAP [10]. A few studies have shown that elevated levels of the tissue necrosing factor-α (TNF‐α) predicted SAP [11,12]. The opinions regarding the utility of the TNF‐α in prediction SAP are contradictory. According to another authors, it is elevated only in a small percentage (9-36%) of SAP patients, and it cannot be used as a prognostic factor in AP [10,13,14]. Similarly, the significance of IL‐10 regarding severity of AP is yet not clear because of the contradictory reports [10,11,15].The other cytokins that have been assessed for predicting AP are as follows: IL-8, and IL-18 [8].

Angiopoietin-2 (Ang-2), a type of glycoprotein acting selectively on endothelial cells and leading to an increase in endothelial permeability, has been identified to be a useful predictor for acute gastrointestinal injury and intestinal barrier dysfunction in patient with AP [16].

    The morphologic severity of AP can be determined using a CT severity index (CTSI) that was developed by Balthazar and then modified and extended to monitor organ failure by Silverman, Banks, and colleagues in 2004 [8].

Concluding, the following cutoff values positively correlate with the severity grade of AP: Ranson ≥ 3, Glasgow ≥ 3, MOSS ≥ 5, BISAP ≥ 2, APACHE II ≥ 8, CTSI ≥ 5, procalcitonin ≥ 0.5 ng/ml, CRP ≥ 150 mg/l, and IL-6 ≥ 50 pg/ml [8].

iii) Multiple factors contribute to the pathophysiology of SAP – REE concept; inflammatory cytokines; protein catabolism; hyperglycemia; micronutrient abnormalities: a section or two describing these is vital to the review. A commentary on the exocrine pancreatic insufficiency and its core link to malnutrition/nutrition deficiency is also an important aspect to introduce nutritional support in patients with SAP.  

Answer: All these factors contributing to the pathophysiology of SAP – REE concept; inflammatory cytokines; protein catabolism; hyperglycemia; micronutrient abnormalities as well as a commentary on the exocrine pancreatic insufficiency and its core link to malnutrition/nutrition deficiency have been presented in the introduction as follows:

Premature intraacinar activation of trypsinogen leads to the cytokine storm in AP. Therefore, various cytokines have been tested to be useful for prediction SAP. Among, them, proinflammatory interleukin-6 (Il-6) is the most useful parameters. It has been confirmed that the plasma level of IL‐6 is elevated early phase of AP. Therefore, it is a sensitive and specific marker to predict organ failure and SAP [10]. A few studies have shown that elevated levels of the tissue necrosing factor-α (TNF‐α) predicted SAP [11,12]. The opinions regarding the utility of the TNF‐α in prediction SAP are contradictory. According to another authors, it is elevated only in a small percentage (9-36%) of SAP patients, and it cannot be used as a prognostic factor in AP [10,13,14]. Similarly, the significance of IL‐10 regarding severity of AP is yet not clear because of the contradictory reports [10,11,15].The other cytokins that have been assessed for predicting AP are as follows: IL-8, and IL-18 [8].

Angiopoietin-2 (Ang-2), a type of glycoprotein acting selectively on endothelial cells and leading to an increase in endothelial permeability, has been identified to be a useful predictor for acute gastrointestinal injury and intestinal barrier dysfunction in patient with AP [16].

  • Disturbances of the nutritional status in SAP

SAP is a serious acute disease. In these patients, resting energy expenditure (REE) increases due to inflammation-induced hypermetabolism and/or septic complications. In SAP, protein catabolism and increased energy requirements are observed. The disease leads to undernutrition and disturbances of water / electrolyte and acid-base balance. Therefore, energy and protein requirements are higher in patients with SAP  [17,18,19]. The pancreatic damage and infectious complications are associated with hyperglycemia which must be controlled. Hyperglicemia is induced by damage to beta cells, insulin resistance, and infectious complications. Therefore, insulin therapy should be considered in nutritional support in patients with SAP. Acute pancreatitis is usually associated with deficiency of numerous vitamins and micronutrients such as vitamins B1, B2, B3, B12, C, A, folic acid, and zinc. Moreover, dysregulations in water / electrolytes and of acid‐base balance are noted in SAP. Hypocalcemia is observed in 40-60% of patients due to saponification of calcium. The other disturbances are as follows: hypomagnesemia, decreased parathyroid hormone release, and increased calcitonin levels, hypertriglicerydemia [19]. The level of triglicerydes must be closely monitored during parenteral nutrition including lipid emulsions. It has been noted that a great protein loss with a negative nitrogen balance occurred in 80% of patients with SAP [3]. The nitrogen loss can be up to 20–40 g/d. Negative nitrogen balance is associated with increased mortality [20].

The role of supplementation of pancreatic enzymes is different in AP and CP patients. In patients with AP, generally these supplements are not recommened except obvious pancreatic exocrine insufficiency (PEI). The informations regarding occurence of PEI in AP patients during hospitalization are not sufficient. PEI can be caused by the pancreatic necrosis, but it is manifested following AP [24]. The knowledge of PEI after AP is better. PEI is noted at least during the first 6 to 18 months after AP, and the grade of dysfunction is related to the AP severity [25]. PEI is common in patients with CP. The low-fat oral diet in MAP, or special formula of enteral/parenteral nutrition in SAP (when oral diet is not possible) are commonly used in most patients. In CP, supplementation of exogenous pancreatic enzymes is very essential due to maldigestion and malabsorption in the disease course [26]. A large meta-analysis by Hollemans et al [27], containing 32 studies (1495 patients), showed the 27.1% prevalence of PEI after AP. It was positively correlated with degree of pancreatic parenchymal injury, alcoholic cause (22.7%), SAP (33.4%), and necrotizing pancreatitis (32%). Therefore, clinical observations in terms of symptoms of maldigestion (diarrhea, steatorrhea) and/or non-invasive pancreatic function investigations (such as fecal fat and fecal elastase) for at least 6-18 months after AP, especially in alcoholic, severe, and necrotizing pancreatitis are recommended. In clinically relevant PEI, supplementation of pancreatic enzymes is recommended [19,27].

  1. iv) Section 1, page 2 – the last few sentences: I am assuming the purpose of this review is not to “resolve” long-standing questions in the field or perform a meta-analysis but present readers with the latest information in context of existing literature within the field. I don’t think the authors can ever “resolve” any fundamental questions just by performing a literature search. Please change language to make clearer to the readers the purpose of this review.

Answer: Our language has been changed as follows:

Some issues regarding the optimal timing, type (enteral nutrition (EN) vs parenteral nutrition (PN), via nasogastric tube (NGT) or nasojejunal tube (NJT)), and the role of immunonutrients, such as glutamine (Gln), Arginine (Arg), polyunsaturated omega-3-fatty acids (PUFA), nucleotids in nutrition of patients with SAP, are discussed in this review.

  1. It is beyond my understanding as to why the authors want to fill pages - There is no value added information in section 3 (ESPEN guidelines). Only one reference (reference 6) has been listed out which I am certain, all readers can read by themselves. The information is irrelevant and rather poorly presented in my opinion. Summarizing the guidelines, and in context of what is known in literature is very important (14 guidelines itself published (within 2004-2008), Greenberg et al, Canadian Journal of Surgery, 2016). What are some of the disagreements over timings and types of nutritional intervention? How does the American Gastroenterology Association/ UK guidelines compare to the ESPEN – highlighting the similarities/dissimilarities would be some key points that would make an interesting read in this section in my opinion.

Answer: The comparison of ESPEN, the American Gastroenterology Association and UK guidelines, according to the Reviewer’s suggestion, has been presented in section 3 as follows:

  1. European Society for Clinical Nutrition and Metabolism (ESPEN), Americal Gastroenterological Association (AGA), and UK guidelines on clinical nutrition in severe acute pancreatitis

          The ESPEN, AGA, and UK guidelines recommend the early oral nutrition in MAP patients and enteral nutrition in SAP patients with impossibility of oral feeding. Parenteral nutrition should be reserved for patients in whom enteral  nutrition is not possible or not tolerated. According to ESPEN guidelines, nasogastric tube is preferred rather than nasojejunal tube, and nasojejunal tube should be reserved for patients with gastric otlet obstruction. Also, according to UK guidelines, enteral feeding via nasogastric tube should be effective in 80% of patients. AGA recommends to use either nasogastric or nasojejunal tube. Regarding optimal timing of starting nutrition, ESPEN guidelines recommend starting nutrition within 24-72 hours of admission. According to AGA, early feeding is not successful in all AP patients due to pain, vomiting, or ileus, and feeding may need to be delayed beyond 24 hours in some cases [26,28,29].

The ESPEN authors recommend the use of parenteral glutamine in PN and they do not recommend immunonutrition in other cases in SAP. There is no information regarding the use of glutamine and immunonutrition in AGA recommendations. According to UK guidelines, there is no sufficient reports to recommend standard immune enhanced formulations. The probiotics and a supplementation of pancreatic enzymes are not recommended by ESPEN. There is no information on probiotics and pancreatic enzyme supplementation in AGA and UK guidelines [26,28,29,30].

  1. Section 5 – EN vs PN: The authors have a very dry manner of presenting information. All the literature search in section 5 can be very beautifully tabulated and summarized into columns –“author”/”findings”/”type of analysis”/”outcomes”. Visual presentation of this table will add a lot of value to the paper and help readers assimilate important information very easily/quickly. Also, why have the authors mixed up EN vs PN data for AP and SAP when the paper specifically talks about nutrition management in SAP?

Why is this statement in section 5 when the entire section 5 is about EN/PN? -  “Ma et al [24], in RCT, demonstrated that nasogastric tube feeding did not influence on dysmotility symptoms in AP patients although the appetite increased

Answer: The table summarizing all the presented articles has been added. Some of the presented studies include the analysis of all degrees of AP (from mild to severe acute pancreatitis), and therefore are discussed in this review. This information has been added in the manuscript and the table. Conclusions and our viewpoint have been presented. The part of the text   “Ma et al [24], in RCT, demonstrated that nasogastric tube feeding did not influence on dysmotility symptoms in AP patients although the appetite increased” has been primarily presented in regards to the impact of enteral nutrition via nasogastric tube on dysmotility of the alimentary tract in AP patients. This article does not directly regards comparison between enteral and parental nutrition. According to the Reviewer’s suggestion it has been removed.

5.The optimal route of nutritional support in SAP patients: enteral versus parenteral nutrition

As we mentioned above, it has been shown that the enteral route is optimal for NS in SAP patients. Apart mentioned above advantages, EN is safe and less expensive compared to PN [37]. There are numerous reports which confirm this theory in the world literature.

A 2008 meta-analysis (including  6 RCTs involving 224 patients) showed that EN was associated with a significantly (p<0.05) lower rate of infectious complications, local complications, organ failure, MODS and mortality compared to total parenteral nutrition (TPN). The authors did not find significant (p>0.05) differences in nutrition related complications, and other non-AP-related complications between patients received EN and TPN. Thus, EN was safer compared to TPN in SAP patients [38].

A later meta-analysis (including 8 trials, 5 trials regarding SAP, involving 348 patients) revealed than EN was associated with a significantly (p<0.05) lower rate of mortality, MODS, septic complications, and indications for surgery compared to patients receiving TPN. Additionally, the authors observed the trend towards decreased duration of hospitalization. They also suggested that EN should be standard nutritional intervention in AP patients [32].

A meta-analysis by Yi et al including 8 RCTs, 381 patients  showed a significantly lower mortality, infections, MODS, and surgery rates in patients receiving EN compared to PN. Duration of hospitalization as well as duration of nutrition were comparable in the both groups [33].

A 2018 meta-analysis involving 9 RCTs (500 patients) revealed a significantly lower  rate of mortality, complications, MODS, and surgery, as well as shorter duration of hospitalization in EN patients compared to those who received PN. Based on these results, the authors recommended EN as the preferred nutrition in SAP patients [39].

Another 2018 meta-analysis involving 11 RCTs (562 patients) showed a significantly lower rate of mortality and infectious complications as well as shorter duration of hospitalization in the EN group compared to the PN group. MODS rate was comparable in the both groups [40].

A significant lower rate of mortality and MODS in patients receiving EN compared to patients receiving PN was reported in a meta-analysis by Yao et al involving 348 critically ill patients (5 RCTs) [34].

The retrospective analysis by Tao et al showed a significantly lower rate of infectious complications, MODS, and mortality as well as shorter total duration of hospitalization and duration of hospitalization in the Intensive Care Unit (ICU) in patients receiving EN. In patients receiving PN, a significantly higher hyperglycemia was reported  [41].

The other study showed a respiratory failure (n=3) and non-respiratory single organ failure (n=3) only in the PN group, and the shorter duration of hospitalization and lower cost of hospitalization in patients receiving EN [42].

It is known that gastrointestinal dysmotility can lead to EN intolerance in SAP patients. It is also known that EN increases gastrointestinal motility and has got the trophic impact on enterocytes [43]. Therefore, it is important to continue EN in SAP despite the gut dysmotility secondary to abdominal compartment syndrome (ACS) due to ascites [44]. Hongyin et al [44] analyzed the impact of abdominal paracentesis drainage (APD) on possibility of enteral feeding in AP patients. This study included 161 AP patients hospitalized between January 2015 and April 2016. The patients were divided into two groups: the APD group and the non-APD group. This study showed that APD might enable the supply of enteral feeding in AP patients [44].

Conclusions. All presented articles have shown the superiority of EN in SAP patients. According to most authors, EE is associated with a lower morbidity and mortality rates, shorter duration and lower cost of hospitalization. Summary of articles regarding comparison of EE and PN in SAP is presented in Table 2.

Our viewpoint. We fully agree with these opinions, and also prefer EN in nutritional support in SAP patients. The all above mentioned articles were summarized in Table 2.

                                                                       Table 2. Summary of articles regarding comparison of enteral and parenteral nutrition in SAP.

Author

Findings

Type of analysis

Outcomes

Cao et al [38]

Lower rate of infectious complications, local complications, organ failure, MODS and mortality in EN

Complication rates comparable

Meta-analysis including

6 RCTs (224 patients) Comparison of EN and PN in SAP

EN safer compared to PN in SAP patients

Al-Omran et al [32]

Lower rate of mortality, MODS, septic complications, and indications for surgery than in EN

Meta-analysis including

8 (5 regarding SAP) RCTs (348 patients)

Comparison of EN and PN in AP

EN should be standard nutritional intervention in AP

Yi et al [33]

Lower mortality, infections, MODS, and surgery rates in EE.

Comparable duration of hospitalization and nutrition

Meta-analysis including

8 RCTs (381 patients)

Comparison of EN and PN in SAP

EE superior to TPN in SAP patients

Li et al [39]

Lower  rate of mortality, complications,MODS, and surgery, shorter duration of hospitalization in EN

Meta-analysis including

9 RCTs (500 patients)

Comparison of EN and PN in SAP

EE is preferred rather than TPN in SAP patients

Wu et al [40]

Lower rate of mortality and infectious complicationsshorter duration of hospitalization in EN. Comparable MODS rate in EE and PN.

Meta-analysis including

11 RCTs (562 patients) (348 patients)

Comparison of EN and PN in SAP

EN recommended as an initial treatment for patients with SAP

Yao et al [34]

Lower rate of mortality and MODS in EE.

Meta-analysis including 5 RCTs (348 patients)

Comparison of EN and PN in SAP

EN should be recommended as the preferred route of nutrition for critically ill patients with SAP

Tao et al [41]

Lower rate of infectious complications,MODS, and mortality, shorter total duration of hospitalization and duration of hospitalization in the ICU.

Retrospective

analysis of 185 patients

Comparison of EE and PN in SAP

EE superior to PN in SAP patients

Gupta et al [42]

Lower rate of respiratory and non-respiratory organ failure, shorter duration of hospitalization, lower cost of hospitalization in EE

Randomized controlled trial (17 patients)

Comparison of EE and PN in SAP

EE safer and less expensive tha PN in SAP patients

SAP, severe acute pancreatitis; EN, enteral nutrition; PN, parenteral nutrition; RCT, randomized controlled trial; MODS, multiorgan dysfunction syndrome; ICU, intensive care unit.

  1. Section 6 – Again, the authors have mixed information related to AP and SAP – please decide what literature the authors want to present in this review. While the information is interesting (for SAP), but please present another table (similar to one suggested in point 3 to make information digestible and appealing). What are the authors’ viewpoints? - Information in each section without having insight or a discussion about the subsection being presented can be very dull for the reader. For example – It is known that the timing (before 24h; 48h) or the route (oral/NG/NJ) of early EN differ in different nutrition support guidelines. A commentary on this would be a vital part of presenting information in the review.

Answer: The table summarizing all the presented articles has been added. Some of the presented studies include the analysis of all degrees of AP (from mild to severe acute pancreatitis), and therefore are discussed in this review. This information has been added in the manuscript and the table. Conclusions and our viewpoint have been presented as follows:

  1. The optimal timing of nutritional support in SAP patients

It is known that immune dysregulation secondary to the cytokine storm is observed in SAP patients. Therefore, currently, there is an opinion that early enteral nutrition (EEN) could increase antioxidant activity, modulate inflammatory response, and decrease the risk of MODS [45]. Sun et al [45] assessed the impact of EEN on the immunological function and course in SAP patients. This was a single-center prospective RCT. The patients were divided into two groups: EEN or delayed enteral nutrition (DEN). EN was initiated within 48 hours of admission in EEN group, whereas from the 8th day in DEN group. The CD4+ T-lymphocyte percentage, CD4+/ CD8+ ratio, and the C-reactive protein (CRP) levels were significantly lower in EEN group compared to DEN group. The immunoglobulin G (IgG) levels and human leukocyte antigen-DR expression were significantly higher in EEN patients compared to DEN group. There was no difference in CD8+ T-lymphocyte percentage, IgM and IgA between the two groups. The rate of SIRS, MODS, and pancreatic infection as well as the duration of ICU hospitalization were significantly lower in EEN group compared to DEN group. The hospital mortality rate was comparable in the both groups. The study showed that EEN could improve the course, but did not decrease mortality in SAP patients [45].

In another prospective pilot study involving 60 SAP patients, Sun et al [46] analyzed the impact of EEN on intra-abdominal pressure (IAP) and disease severity in SAP patients. The authors compared two patients‘ groups distinguished depending on the nutrition timing. EN was initiated within 48 hours of admission in EEN group and from the 8th day in DEN group. This study demonstrated that EEN did not increase IAP. Similarly to the above cited study, this trial showed that EEN could decrease AP severity and clinical course, but did not decrease mortality in SAP patients [46].

Zou et al [47] compared EEN (within 72 hours) and DEN (later than 72 hours, within 7 days) in SAP patients hospitalized in the ICU (the median Ranson score: 3). Hospital mortality, duration of hospitalization, percentage of patients who needed mechanical ventilation, surgery or continuous renal replacement therapy, incidence of local and systemic septic complications, acute kidney injury were significantly lower in the EEN group compared to the DEN group. The timing of initiation of enteral feeding was a significant prognostic factor for pancreatic infection. Therefore, in authors‘ opinion,  EEN should be started within 72 hours of SAP onset [47].

A systematic review by Vaughn et al [48] compared early (≤48 hours) and delayed (>48 hours) of admission enteral feeding. This study included the patients with all degrees of AP severity. Four trials involved patients with pSAP. Alhough in patients with mild to moderate pancreatitis, early nutrition was associated with a shorter duration of hospitalization (in 4/7 studies), among patients with SAP, there was no statistically significant difference in outcomes between EEN and DEN. Moreover, there was no adverse effects of early nutrition in this analysis [48].

In another study by Bakker et al [49], the results in AP patients who had received EN < 24 hours (EEN) or >24 hours (DEN) of admission were presented. The study included 8 RCTs involving 165 patients (100 receiving EN, and 65 receiving DEN). Five of 8 trials included patients with pSAP based on APACHE-II, Imrie score, or CRP. In this study, a lower rate of complications was reported in EEN group compared to DEN group [49].

Also, Bakker et al [30] performed a multicenter RCT comparing EEN with the use of nasojejunal tube with an oral diet at 72 hours of admission in SAP patients (severity score according to APACHE II ≥8). This study included 208 patients from 19 Dutch hospitals. The patients were divided into two groups: nasojejunal tube feeding within 24 hours (EEN) and oral nutrition initiated 72 hours of admission (on-demand group), with insertion of the feeding tube in cases of intolerance of the oral diet. Infectious complications and mortality rate were comparable in the two groups [30].

In a retrospective study by Wereszczyńska-Siemiątkowska et al [51], 197 patients with predicted SAP (pSAP) were divided into two groups: EEN (< 48 hours of admission) including 97 patients and DEN (>48 hours) including 100 patients. A significantly higher number of infectious complications, respiratory failure, and hospitalization in the ICU was noted in DEN group compared to EEN one. The MODS and surgical interventions rates were comparable in the both groups. The mortality rate was significantly higher in patients receiving DEN. Moreover, a longer time of initiation of nutritional intervention was associated with a higher risk of local complications such as infected necrosis or fluid collection. In authors’ opinion, EN in SAP should be started within 48 hours of admission [51].

A meta-analysis including 10 RCTs involving 1051 patients confirmed the efficacy and safety of EEN (< 48 hours of admission)  in SAP and pSAP patients [52].

Another meta-analysis including 11 studies containing 775 AP patients (seven articles stratified into the pSAP or SAP subgroup) showed a significant lower rate of overall infectious complications, catheter-related septic complications, and local infectious complications, as well as significantly lower hyperglycemia, shorter length of hospital stay, decreased  mortality in patients receiving EEN. Only, the rates of pulmonary complications were comparable in the both groups. Based on this analysis, the authors concluded that EEN (within 48 hours of admission) was associated with the improved clinical outcomes and a reduction of complications in AP generally as well as in pSAP and SAP [53].

A meta-analysis including 8 studies containing 727 AP patients, comparing the use of EEN, DEN, and PN, showed a significantly lower number of local infectious complications and MODS in pSAP and SAP patients receiving EEN. There were no advantages of EEN in MAP and MSAP patients. Based on these results, the authors recommended EEN only for pSAP and SAP patients (but not for patients with the lower severity degrees of AP) [54].

Conclusions. According to most authors and guidelines, EN should be started within 24-48 hours in SAP or pSAP in order to prevent the gut barier dysfuntion and dysmotility, and infectious complications. Moreover, according to most authors, EEN is also associated with a shorter duration of hospitalization and lower hospital costs [55,56]. Summary of articles regarding comparison of early and and delayed enteral nutition in SAP is presented in Table 3.

Our viewpoint. In our opinion, in pSAP and SAP patients, EEN should be started within 48 hours of admission to hospital. Starting EN within first 24 hours is frequently not possible because of the clinical status (abdominal pain, nausea, vomiting), metabolic disturbances (acid-base balance, dehydratation, electrolyte deficiency) which should be first controlled. Besides, during first 24 hours, diagnostic process is performed to assess the severity of AP. In accordance with above mentioned information, prediction of SAP using some scores (the Ranson, Glasgow scores) requires 48 hours, and for patients with not predicted SAP, EN is not recommended.

                                                                       Table 3. Summary of articles regarding comparison of early and delayed enteral nutrition in SAP.

Author

Findings

Type of analysis

Outcomes

Sun et al [45]

Lower CD4+ T-lymphocyte %, CD4+/ CD8+ ratio, CRP,

Higher IgG and HLA-DR in EEN

Lower SIRS, MODS, and pancreatic infection rates

Lower duration of hospitalization in the ICU in EEN

Randomized controlled trial including 60 patients

Comparison of EEN (48hours) and DEN (8th day) in SAP

EEN improves the course, but not decreases mortality compared to DEN in SAP patients

Sun et al [46]

EEN does not increase IAP

Decreased AP severity and clinical course, but did not decreased mortality in EEN

Randomized controlled trial including 60 patients

Comparison of EEN (48hours) and DEN (8th day): impact on IAP and disease severity in SAP

EEN improves the course, but does not decrease mortality compared to DEN,

EEN does not increase IAP in SAP patients

Zou et al [47]

Lower hospital mortality, duration of hospitalization, % of patients requiring mechanical ventilation, surgery, continuous renal replacement therapy,

Lower incidence of local and systemic septic complications, acute kidney injury EEN

Retrospective analysis of 93 patients

Comparison of EEN (within 72 hours) and DEN (later than 72 hours, within 7 days) in SAP

EEN should be started within 72 hours of SAP onset

Vaughn et al [48]

Systematic review including 11 RCTs (11 RCTs on SAP) (948 patients)

Comparison of EEN (≤48 hours) and DEN (>48 hours) in all severity degress of AP

No difference in outcomes between EEN and DEN in SAP patients

Bakker et al [49]

Lower rate of complications in EEN

Meta-analysis of 8 RCTs (165 patients)

Comparison of EEN (≤24 hours) and DEN (>24 hours) in all severity degress of AP

EEN is associated with a reduction of complications

Bakker et al [50]

Comparable rates of complications and mortality

Multicenter RCT including 208 patients

Comparison of EEN EEN with an oral diet at 72 hours of admission in SAP

EEN is not superior to an oral diet after 77 hours in SAP patients

Wereszczyńska-Siemiątkowska et al [51]

Lower mortality rate,frequency of infected necrosis/fluid collections, respiratory failure, and a need for ICU hospitalization in EEN

Retrospective analysis of 197 patients

Comparison of EEN (≤24 hours) and DEN (>24 hours) in pSAP

EE in SAP should be started within 48 hours after admission to hospital

Song et al [52]

Lower mortality, MOF, surgery, systemic and local infection rates in EEN

Comparable SIRS and other local complication rates in EEN

Meta-analysis including 10 RCTs (1051 patients)

Comparison of EEN (≤48 hours) and DEN (>48 hours) or PN in pSAP, SAP

EEN is efficient and safe in pSAP and SAP patients

Li et al [53]

Lower number of infectious complications, local complications, infected necrosis, morrespiratory failure, mortality rate and hospitalization in the ICU in EEN

Retrospective analysis of 197 patients

Comparison of EEN (≤48 hours) and DEN (>48 hours) in pSAP

EE in SAP should be started within 48 hours after admission to hospital

Song et al [52]

Lower rate of overall infectious, catheter-related septic and local infectious complications lower hyperglycemia, shorter length of hospital stay, decreased  mortality in EEN

Comparable pulmonary complications

Meta-analysis of 11 studies (775 patiens)

Comparison of EEN (≤48 hours) and DEN (>48 hours) in pSAP

EEN improves the outcome and teduces complication rate in pSAP and SAP patients

Qi et al [54]

Lower number of local infectious complications and MODS only in EEN in pSAP and SAP

Meta-analysis including 8 studies (727 patients)

Comparison of EEN (<24 hours) with DEN, PN in with all AP severity degrees

EEN should be used only in pSAP and SAP patients (not lower degrees)

No advantages of EEN in MAP and MSAP patients

SAP, severe acute pancreatitis; pSAP, predicted severe acute pancreatitis; EEN, early enteral nutrition; DEN, delayed enteral nutrition; PN, parenteral nutrition; RCT, randomized controlled trial; MODS, multiorgan dysfunction syndrome; MOF, multiorgan failure; ICU, intensive care unit, IAP, intra-abdominal pressure.

  1. Section 7 – Similar changes as mentioned above. Authors can present a Table 3 with “author recommendations” and summarize the last paragraph of this section within this table.

Answer: The table summarizing all the presented articles has been added. Some of the presented studies include the analysis of all degrees of AP (from mild to severe acute pancreatitis), and therefore are discussed in this review. This information has been added in the manuscript and the table. Conclusions and our viewpoint have been presented as follows:

  1. The nasogastric versus nasojejunal tube in enteral nutrition of SAP patients

Based on above mentioned reports, it is obvious that EEN is superior to PN in AP patients. But what about the enteral rout? Which rout is optimal for SAP patients. Historically, nasojejunal tube (NJT) was preferred in EN in order to minimise pancreatic secretion. According to the current mentioned above theory of „no pancreatic rest“, the nasogastric tube (NGT) might be preferred in SAP patients, because it has been proven that insertion of the feeding tube in the stomach does not increase pancreatic secretion in SAP patients, and theoretically gastric placement of the feeding tube is easier. Thus, an opinion on the optimal rout of EN has been also changed. During 50 years, EE using NJT feeding was considered contraindicated in AP. According to „pancreatic rest“ theory, TPN was used in all SAP patients. Similarly due to this theory, NGT was primarily contraindicated in SAP patients. There are numerous studies comparing NGT and NJT in the literature.

In 2005, Eatock et al [57] performed a pilot RCT comparing NGT and NJT for EN in SAP patients. The authors proposed the use of NGT for EN in SAP patients. Overall mortality was 24.5% including  5/27 (18.5%) in the NGT patients and 7/23 (30.4%) in the NJT patients. This study demonstrated that EN via NGT was easier and equally effective compared to EN via NJT in SAP patients [57].

In a study by Singh et al [58], the infectious complications in the NGT (23.1%) and NJT (35.9%) patients were comparable (p<0.05). Also, an abdominal pain during refeeding, bowel permeability, and endotoxemia were comparable in the two groups. This study revealed that EEN through NGT was not worse, but comparable to EEN through NJT in SAP patients [58].

A meta-analysis by Petrov et al [59] including four studies (92 pSAP patients)  demontrated similar effects in the NGT and NJT groups, including mortality rate and intolerance of EN (p>0.05). Thus, this study showed that NGT was safe and well tolerated in pSAP patients [59].

Another meta-analysis including 3 RCTs (157 pSAP patients) showed a similar mortality rate, and complications such as tracheal aspiration, diarrhea, increased abdominal pain, as well as covering of energy requirement in the both groups. Thus, this study also showed that EN via NGT was not worse than EN via NJT. The authors‘ conslusions were the same as mentioned above [60].

A meta-analysis by Nally et al [61] revealed a similar covering of the energy requirement, tolerance of enteral feeding, an increase of abdominal pain, and tube displacement in the both groups. Vomiting (13.3%) and diarrhea (12.9%) were the most frequently noted during EN via NGT. The other adverse effects were as follows: increased aspiration (9.1%), increased abdominal pain (7.5%), abdominal distension (1.5%), and increased AP severity (1.6%). The study demonstrated 90% sufficiency of EN via NGT feeding [61].

Another recent meta-analysis demonstrated a similar  incidence of mortality, MODS, infectious complications, the tube insertion and enteral feeding related complications, indications for surgery, intolerance of enteral feeding with necessity of PN administration, increased abdominal pain in the two groups [62].

Conclusions. Based on the mentioned above studies, generally EN may be provided by the NGT and NJT in SAP patients. NJT seems to be better than NGT in cases of gastroparesis associated with an aspiration risk, swelling of the pancreas, or large postinflammatory pancreatic cysts impressing the stomach or duodenum. However, insertion of NJT is more complicated, more difficult and frequently it must be performed during endoscopic approach or under fluoroscopic guidance, and may need additional sutures or clips in order to fix its placement [18]. It should be added that longer duration of NGT or NTJ can lead to complications such as discomfort for a patient, dislocation or unintentional tube removal, aspiration, sinusitis, and injury of the nasal cavity. Therefore, for patients requiring enteral feeding for a long time (>30 days), according to general nutritional recommendations, percutaneous gastrostomy or microjejunostomy should be considered. [18]. Summary of articles regarding comparison of NGT and NJT in SAP is presented in Table 4.

Our viewpoint. We prefer enteral feeding via NGT in most patients, and NJT in patients with gastric outlet obstruction, because the insertion of NGT is easier and does not require endoscopic control in contrast to NJT.

Table 4. Summary of articles regarding comparision of nasogastric versus nasojejunal tube in enteral nutition in SAP.

Author

Findings

Type of analysis

Outcomes

Eatock et al [57]

Comparable outcome in NGT and NJT

Mortality (18.5%) in NGT and (30.4%) in NJT patients

Pilot Randomized control trial including 50 patients

Comparison of NGT and NJT in EE in SAP

EN via NGT was easier and equally effective compared to EN via NJT in SAP patients

Singh et al [58]

Comparable rate of infectious complications,abdominal pain during refeeding, bowel permeability, and endotoxemia in both groups

Randomized control trial including 78 patients

Comparison of NGT and NJT in EE in SAP

EE via NGT comparable to EE via NJT in SAP patients

Petrov et al [59]

Comparable effects including mortality and feeding intolerance in both groups

Meta-analysis including 4 trials (92 patients)

Comparison of NGT and NJT in EE in pSAP

EE via NGT safe and well tolerated in pSAP patients

Chang et al [60]

Comparable mortality, and complications (tracheal aspiration, diarrhea, increased abdominal pain), covering of energy requirement in both groups

Meta-analysis including 3 trials (157 patients)

Comparison of NGT and NJT in EE in pSAP

EE via NGT safe and well tolerated in pSAP patients

Nally et al [61]

Comparable covering of the energy requirement, tolerance of enteral feeding, increase of abdominal pain, tube displacement were similar in both groups

Meta-analysis including 4 RCT

Comparison of NGT and NJT in EE in SAP

NGT feeding is efficacious in 90 % of SAP patients

Dutta et al [62]

Comparable mortality, MODS, infectious complications, tube insertion and enteral feeding related complications, indications for surgery, intolerance of enteral feeding with necessity of PN administration, increased abdominal pain in both groups

Meta-analysis including 5 RCT (220 patients)

Comparison of NGT and NJT in EE in SAP

Insufficient evidence regarding superiority/inferiority/equivalence between NGT and NJT in EE in SAP patients

SAP, severe acute pancreatitis; pSAP, predicted severe acute pancreatitis; EN, enteral nutrition; RCT, randomized controlled trial; MODS, multiorgan dysfunction syndrome.

  1. Section 8 – Again, the authors have a very unusual style of presenting information in their paper as I reach section 8. While I understand and am sensitive to the fact that English may not be their native language, the authors can take vital cues on how to present data from some excellent papers they have referenced for their review. I would expect the authors to completely restructure their style and information presentation for the entire review. The authors do not need to start each sentence with the author name and paper they are referencing and then summarize the work. Making crisp short sentences that impart the relevant information followed by the reference number in the end to validate the statement would suffice in my opinion.

Answer: Our style of presenting information has been changed in the whole manuscript according to the Reviewer’s suggestions. In addition, the table summarizing all the cited articles has been presented in section 8 as follows:

  1. Composition of enteral nutrition formulas in SAP patiens

          Historically, it has been thought that elemental and semi-elemental formulas less stimulate pancreatic secretion, are associated with a lower digestion, and are easily absorbed within a small bowel. Currently, the use of polymeric formulas can be sufficient and useful in SAP patients [19]. In a randomized prospective pilot study comparing a semi-elemental formula with a polymeric formula in enteral feeding in AP patients, tolerance of the both formulas was good. In the both groups, steatorrhea and creatorrhea were lower than normal. A significantly shorter duration of hospitalization and a lower loss of weight (p<0.05) were noted in the patients receiving a semi-elemental formula. Therefore, this study showed comparable food tolerance in the both groups, but better clinical outcome in the patients receiving a semi-elemental formula [63].

A meta-analysis including 20 RCTs ( 1070 patients, involving 825 patients with SAP) showed no associations between enteral feeding intolerance and a kind of enteral formula (semi-elemental or polymeric formula), the probiotics administration, and immunomodulating nutrition. The infectious complications and mortality rates were similar in compared groups. Regarding formula composition, the authors concluded that administration of both polymeric and semi-elemental formulas was associated with a similar risk of feeding intolerance, infectious complications or mortality in AP patients [64].

A retrospective study including 948 patients (382 patients receiving the elemental formula and 566 patients in the control group) demonstrated a similar incidence of mortality (10.2% vs. 11.0%), sepsis (5.0% vs. 7.1%), hospital-free duration (54 days vs. 51 days), and total health-care costs [$29,360 vs. $34,214) in the two groups. Thus, this large study showed comparable results of enteral feeding with the use elemental, semi-elemental and polymeric formulas in AP patients. This study involved  817 patients with SAP [65].

          Reviewing the literature, we have found a study on the association between a polymeric formula in EEN and chylous ascites (CA) in SAP patients. Zhang et al [66] described CA in SAP patients receiving EEN with a polymeric formula. This retrospective study included SAP 85 patients. The SAP patients were divided into two groups according to timing of EN introduction: EEN (<72 hours) and  DEN  (>72 hours). The chylous ascites was noted in 13 (15.29%) of 85 patients. CA was more frequently reported in patients receiving EEN patients with the use of a polymeric formula (9/33, p < 0.05). Duration of hospitalization in the ICU and in mortality rate were comparable regardless the CA presence. The study demonstrated a higher CA incidence in patients receiving EEN with the use of a polymeric formula in SAP patients, but further studies are needed to confirm these observations [66].

Conclusions. According to most authors, polymeric and elemental formulas are comparable regarding the nutrition tolerance and impact on clinical outcome in SAP patients. Summary of articles regarding composition of enteral formulas in SAP is presented in Table 5.

Our viewpoint. We agree with these observations.

Table 5. Summary of articles regarding composition of enteral formulas in SAP.

Author

Findings

Type of analysis

Outcomes

Tiengou et al [63]

Comparable feeding tolerance in both groups

Shorter duration of hospitalization, lower loss of weight in patients receiving a semi-elemental formula.

Randomized controlled trials including 30 patients

Comparison semi-elemental and polymeric formula in AP patients stratified according to severity

Comparable food tolerance in both groups, but better clinical outcome in patients receiving a semi-elemental formula

Petrov et al [64]

Comparable feeding tolerance,

infectious complications and mortality rates in both groups

Meta-analysis including  trials (1070 patients)

Comparison of semi-elemental or polymeric formula

Comparison of semi-elemental and polymeric formula

The use of polymeric formula, compared to semi-elemental formula, does not lead to increased feeding intolerance, infectious complications or mortality

Endo et al [65]

Comparable mortality, sepsis rates, hospital-free duration, total health-care costs in both groups.

Retrospective cohort study including 382 patients

Comparison of elemental orcontrol formula

Comparable results of EN with the use elemental, semi-elemental and polymeric formulas

SAP, severe acute pancreatitis; pSAP, predicted severe acute pancreatitis; EN, enteral nutrition; RCT, randomized controlled trial; MODS, multiorgan dysfunction syndrome.

  1. Section 9 – IN info is detailed and focuses on SAP but again, another table with summary of information can be very helpful for the reader.

Answer: The table summarizing all the presented articles has been added as follows:

Table 6. Summary of articles regarding immunonutrition and other supplements in SAP.

Author

Findings

Type of analysis

Outcomes

Enteral immunonutrition

Petrov et al [69]

Comparable risk of infectious complications and mortality, duration of hospitalization in both groups

A meta-analysis including 3 RCTs (78 patients)

Comparison of IN nad standard enteral formula in AP patients (from MAP to SAP)

No benefits of IN in EE in AP patients (including SAP patients_

Poropat et al [70]

Comparable overall mortality and SIRS rate in both groups

A meta-analysis including 3 RCTs (78 patients)

Comparison of IN nad standard enteral formula in AP patients (from MAP to SAP)

No benefits of IN in AP patients

Pearce et al [71]

Comparable decreased CRP in both groups

Randomized controlled trial including 31 patients

Comparison of EIN and control feeding in pSAP patients

The cause of the unexpectedly higher CRP values in the study group is unclear

Huang et al [72]

Comparable APACHE II score, duration of hospitalization, costs in both groups

Randomized controlled trial including 32 patients

Comparison of EIN and control feeding in pSAP patients

EIN (Gln, Arg) improves the gut barrier function by reducing the gastrointestinal permeability and decreasing plasma endotoxin level in the early SAP phase 

Singh et al [73]

Comparable infectious complications, prealbumin level, total duration of hospitalization / duration of hospitalization in ICU, and mortality in both groups

Randomized controlled trial including 80 patients

Comparison of EIN (Gln) and control feeding in pSAP patients

No significant impact of Gln on gut permeability in SAP patients

Arutla et al [74]

Comparable rated of infected necrosis and in-hospital mortality in both groups

Higher increase of serum Gln, lower polyethylene glycol, higher decrease of Il-6 in Gln group

Randomized controlled trial including 40 patients

Comparison of standard nutrition and standard nutrition supplemented with enteral Gln in SAP and pSAP patients

Enteral Gln supplementation improves the gut permeability and oxidative stress in SAP and pSAP patients

Parenteral immunonutrition

Jafari et al [75]

Lower mortality rate,

shorter duration of hospitalization PIN group

Meta-analysis including 7 RCTS on PIN supplemented with Gln and / or PUFA

PIN (Gln, PUFA) can improve prognoses in patients with AP

Fuentes-Orozco et al [80]

Increased IL-10, total lymphocyte and lymphocyte subpopulation counts, and albumin levels, improvement of nitrogen balance, lower rate of infectious complications in Gln group

Comparable duration of hospitalization and mortality rate in both groups

Randomized controlled trial including 44 patients

Comparison of standard PN (n = 22) and Gln-supplemented PN in SAP patients

PIN (Gln) may decrease infectious morbidity rate

Xu et al [81]

Shorter duration of acute respiratory distress syndrome, renal insufficiency, acute hepatitis, shock, encephalopathy, and paralytic ileus, and hospitalization, lower APACHE II score, lower infection, surgery and mortality rates in early group

Randomized controlled trial including 80 patients

Comparison of 2 groups of intravenous Gln (early treatment group) or 5 d after (late treatment group) admission in SAP patients

Early Gln supplementation superior to to delayed one in SAP patients

Wang et al [82]

Higher eicosapentaenoic acid (EPA), lower CRP level, better oxygenation index, shorter duration of continuous renal replacement therapy in PUFA group

Randomized controlled trial including 40 patients

Comparison of standard PN and PN supplemented with omega-3-fatty acids

PN supplemented with PUFA diminished the hyperinflammatory response by the EPA increase and the proinflammatory cytokine decrease in SAP patients

Probiotics

Gou et al [84]

No impact of probiotics on pancreatic infection, total infections, operation, mortality rates, duration of hospitalization

Meta-analysis including 6 trials (536 patients)

Analysis of advantages and disadvantages of probiotics on the outcome in pSAP patients

No sufficient data to draw conclusions on the role of probiotics in nutrition in pSAP patients

Besselink et al [85]

Higher infectious complications, mortality, bowel ischemia rates in probiotics group

Multicenter randomized controlled trial including 298 pSAP patients

Comparison of probiotic sand placebo groups

Probiotics do not decrease a risk of septic complications in pSAP patients

Use the probiotic prophylaxis is not recommended in SAP patients

Wang et al [86]

Lowest pancreatic infectious complications, MODS, mortality rate, TNF-α and IL-6 levels, highest Il-10 as well as APACHE II scores in EN + EcoIN

Randomized controlled trial including 183 SAP patients Comparison of receiving PN, EN, or EN + EcoIN

Combination of EcoIN with EN has got more advantages compared to exclusive EN in SAP patients

SAP, severe acute pancreatitis; pSAP, predicted severe acute pancreatitis; EN, early enteral nutrition; PIN, parenteral immunonutrition; RCT, randomized controlled trial; MODS, multiorgan dysfunction syndrome; ICU, intensive care unit.

  1. Section 10 – Summarize your nutritional recommendations with a table/flowchart (for example – like one in Ramanathan and Aadam, Nutrition in Clinical Practice 2019).

Answer: The table summarizing our recommendations and figure with algorithm of nutritional management in SAP based on the literature review have been presented as follows:

Table 7. Summary of our recommendations on nutritional support in SAP based on the literature review.

Aspect of nutritional support in SAP

Our recommendations

The optimal route of feeding

EN is feeding of choice in SAP patients in whom oral nutrition is impossible

Parenteral nutrition is reserved for patients with intolerance or impossibility of EE

The optimal timing of nutrition

EEN (<48 hours of admission) is superior to DEN

EN should be started within 48 hours of admission

NGT versus NJT

NGT is the route of choice

NJT is preferred in patients with GOO

Immunonutrition

IN supplementation (including Gln in dose 0.3-0.5 g/kg/d) is recommended in PN

Probiotics

Not recommended

SAP, severe acute pancreatitis; EE, enteral nutrition; EEN, early enteral nutrition; DEN, delayed enteral nutrition; PN, parenteral nutrition; NGT, nasogastric tube; NJT, nasojejunal tube; GOO, gastric outlet obstruction; IN, immunonutrition; Gln, glutamine.

Figure 1. Algorithm of nutritional management in SAP.

  1. All sections lack insightful clinical relevance. For example – relapse of GI issues after EN is common; how the nutritional parameters are assessed in relation to what is identified in the imaging modalities?; What are the clinical recommendations to manage nutrient requirements? Pancreatic enzyme secretion by enteral nutrients may result in autodigestion – what is the role of antisecretory agents? All these aspects need consideration in context of the topic being discussed for review.

Answer: Clinical considerations regarding assessment of the nutritional status and nutrient requirements, practical aspects of enteral feeding, complications, and interruption, care and monitoring during enteral and parenteral nutrition as well as the role of antisecretory agents have been presented in the two additional sections (10. Antisecretory management, and 12. The other clinical considerations and practical tips regarding nutritional support in patients with SAP) as follows:

  1. Antisecretory management

Some authors recommend the use of somatostatin, a hormone suppressing the pancreatic secretion in SAP patients. It is administered to rest by inhibiting pancreatic secretions and to prevent autodigestion stimulated by enteral nutrition. In practice, octreotide is used because of its longer half‐life compared with somatostatin (72–98 vs 2–3 minutes) and simpler administration (intravenous continuous infusion vs subcutaneous three doses per day) [20].

Our viewpoint. The data regarding the use of somatostatin/octreotide are not sufficient in order to recommend their standard use in SAP. We recommend intravenous administration of inhibitors of protein pomp (IPP) due to the higher risk of stress ulcer in SAP.

  1. The other clinical considerations and practical tips regarding nutritional support in patients with SAP

Nutritional support should be individually composed. The nutritional risk is assessed according to commonly used nutritional risk score such as Subjective Global Assessment (SGA) and Nutritional Risk Score 2002 (NRS 2002) [26]. The energy requirement should be estimated using indirect calorymetry (IC) if possible, or should be calculated by formula 25-35 kcal/kg/d. The formulas of enteral and parenteral nutrition are composed based on the following nutritional requirements: protein 1.2-1.5 g/kg/d, carbohydrates 3–6 g/kg/day corresponding to blood glucose concentration (aim:<10 mmol/l), lipids up to 2 g/kg/day corresponding to blood triglyceride concentration (aim:<12 mmol/l), Natrium 1–2 mmol/kg/d, potassium 1–2 mmol/kg/d, chlorine 2–4 mmol/kg/d, phosphorus 0,1–0,5 mmol/kg/d, magnessium 0,1–0,2 mmol/kg/d, and calcium 0,1 mmol/kg/d. They sould be modified depending on serum concentrations, metabolic status and balance [87].

Formulations used in EE nad PN should contain proteins, carbohydrates, and fats. In complete PN, solutions of vitamins and micronutrients should be administered. EE via NGT is administered in interrupted boluses (200-300 ml 5-6 times per day under control of gastric residual volume (GRV)) or continuous infusion (30-50 ml/h), while EE via NJT is administered by continuous infusions. The flow velocity should increase gradually: from 20-30 ml/h to 100-125 ml/h. In order to avoid complications (regurgitation, aspiration, or pneumonia), EN via NGT should be interrupted in GRV>200 ml. EE should cover minimally 60% of energy requirement. When it is not possible or in intolerance of EE, PN should be added. PN should be started in volume of 50% of estimated nutritional requirement on day 1, 75% - on day 2, and 100% - from day 3. The hemodynamic status should be controlled and all disturbances/deficiencies of water/electrolites and acid-base balance should be compensated before starting nutrition to avoid re-feeding syndrome. In cases of EE intolerance such as diarrhea, the velocity of feeding should be decreased. When it is not sufficient, PN administration of PN should be considered. The assessment of nutrional requirement and control laboratory investigations should be performed minimally once a week for optimal nutritional support and modification of the type or formula if it is indicated. Also, a care of the tube (in EE) or catheter (in PN) is very important to avoid infectious and other catheter and tube related complications [88].

  1. Minor but important – language and presentation style (narrative flow) requires polishing throughout the draft. The review is very dry to read – lack of tables/figures - makes it very difficult for the reader to sift through so much information.

Answer: Language and presentation style have been changed and improved according to the Reviewer’s suggestion. The seven tables and one figure have been presented.

Round 2

Reviewer 2 Report

The substantial changes and efforts made by the authors are appreciated. Please make some minor corrections/ changes listed below that will help in further refining the review manuscript (please see all changes reflected in the “.pdf” file) -

  1. Please make sure references are in correct order – for example, two references are merged in reference [23]. Please revise to-

[23] – O’Brien and Omer et al. Nutrition in Clinical Practice, 2019. Pubmed ID: 31535736

[24] – Narayanan et al. Nutrition in Clinical Practice, 2020. Pubmed ID: 33002260

  1. Excellent effort to include a total of 7 tables, but a lot of information in the text becomes redundant. Please see the suggested changes as “edited/recommended” in the pdf file.
  2. While it is excellent that the authors present an algorithm, please ensure that it can be clearly see once printed (it is highly pixelated now).
  3. Please also see some “text and grammar” changes as suggested in the .pdf file.

Author Response

Dear Editor,

Dear Reviewer,

Thank you for peer reviewing of our manuscript  nutrients-1156590, entitled " Nutritional Support in Patients with Severe Acute Pancreatitis - Current Standards".

Thank you for your valuable comments and recommendations. We have fully addressed all the comments and my responses appear below. Our revised work includes corrections according to reviewers’ comments in the text. The changes, made according to reviewers’ comments, are highlighted in green print in the text.

We take this opportunity to express my gratitude to the reviewers for their constructive and useful remarks. Their comments allowed us to identify areas in my manuscript that needed modification.

We also thank you for allowing me to resubmit a revised copy of the manuscript.

We hope that the revised manuscript is now acceptable for publication in Nutrients.

Yours sincerely,

Beata Jabłońska.

Responses to Reviewer 2.

  1. Please make sure references are in correct order – for example, two references are merged in reference [23]. Please revise to-

[23] – O’Brien and Omer et al. Nutrition in Clinical Practice, 2019. Pubmed ID: 31535736

[24] – Narayanan et al. Nutrition in Clinical Practice, 2020. Pubmed ID: 33002260

Answer: These two references have been revised, separated, and numbers have been changed as follows:

  1. O'Brien SJ, Omer E. Chronic Pancreatitis and Nutrition Therapy. Nutr Clin Pract. 2019 Oct;34 Suppl 1:S13-S26. doi: 10.1002/ncp.10379. PMID: 31535736.
  2. Narayanan S, Bhutiani N, Adamson DT, Jones CM. Pancreatectomy, Islet Cell Transplantation, and Nutrition Considerations. Nutr Clin Pract. 2020 Oct 1. doi: 10.1002/ncp.10578. Epub ahead of print. PMID: 33002260.

The other reference numbers have been changed.

  1. Excellent effort to include a total of 7 tables, but a lot of information in the text becomes redundant. Please see the suggested changes as “edited/recommended” in the pdf file.

Answer: The redundant informations in the text have been deleted according to the Reviewer’s suggestions.

  1. While it is excellent that the authors present an algorithm, please ensure that it can be clearly see once printed (it is highly pixelated now).

Answer: The file of Figure 1 has been improved.

  1. Please also see some “text and grammar” changes as suggested in the .pdf file.

Answer: „Text” and „grammar” changes according to the Reviewer’s suggestions have been made.
